# Classification Diffusion Models: Revitalizing Density Ratio Estimation

**Shahar Yadin    Noam Elata    Tomer Michaeli**
Faculty of Electrical and Computer Engineering
Technion - Israel Institute of Technology
{shahar.yadin@campus,noamelata@campus,tomer.m@ee}.technion.ac.il

## Abstract

A prominent family of methods for learning data distributions relies on density ratio estimation (DRE), where a model is trained to *classify* between data samples and samples from some reference distribution. DRE-based models can directly output the likelihood for any given input, a highly desired property that is lacking in most generative techniques. Nevertheless, to date, DRE methods have failed in accurately capturing the distributions of complex high-dimensional data, like images, and have thus been drawing reduced research attention in recent years. In this work we present *classification diffusion models* (CDMs), a DRE-based generative method that adopts the formalism of denoising diffusion models (DDMs) while making use of a classifier that predicts the level of noise added to a clean signal. Our method is based on an analytical connection that we derive between the MSE-optimal denoiser for removing white Gaussian noise and the cross-entropy-optimal classifier for predicting the noise level. Our method is the first DRE-based technique that can successfully generate images beyond the MNIST dataset. Furthermore, it can output the likelihood of any input in a single forward pass, achieving state-of-the-art negative log likelihood (NLL) among methods with this property. Code is available on the project's webpage.

## 1   Introduction

A classical family of methods for learning data distributions relies on the concept of density-ratio estimation (DRE) [46]. DRE techniques transform the unsupervised task of learning the distribution of data into the supervised task of learning to classify between data samples and samples from some reference distribution [15, 4, 35, 7]. These methods have attracted significant research efforts over the years [27, 14, 35, 47], particularly for their inherent capability to directly output the likelihood for any given input. However, to date, they have not succeeded in capturing the distribution of complex high-dimensional data, like natural images. Instead, their generative performance was demonstrated only on low-dimensional toy examples and on the simple MNIST handwritten digits dataset [28, 35, 7]. As illustrated in Fig. 1, while the state-of-the-art DRE method, telescoping density-ratio estimation (TRE) [35], succeeds in capturing the distribution of the MNIST dataset [28], it fails on the slightly more complex CIFAR-10 dataset [26].

As opposed to DRE methods, denoising diffusion models (DDMs) [38, 19] have had unprecedented success in generative modeling of complex high-dimensional data, including images [9, 36], audio [25, 5], and video [12, 1]. This has made them perhaps the most prominent technique for learning data distributions in recent years, with applications in solving inverse problems [21, 37], image editing [33, 16, 3, 20] and medical data enhancement [44, 8], to name just a few. However, assessing the likelihood of data samples is a challenging task with DDMs; it requires many neural function

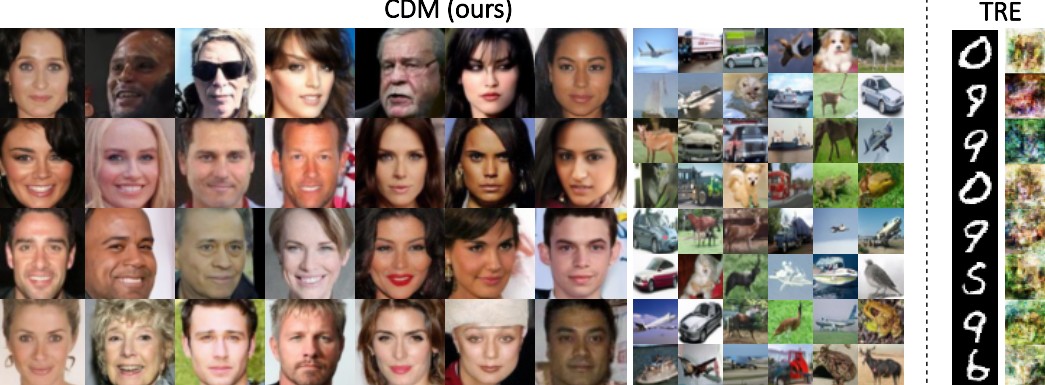

CDM (ours)                                    TRE

Figure 1: **Samples from CDMs (left) trained on CelebA** $64 \times 64$ **and on CIFAR-10, compared to samples from TRE models [35] (right) trained on MNIST and CIFAR-10.** To date, DRE methods have failed to capture the distributions of complex, high-dimensional data, and have been demonstrated only on toy examples or on the simple MNIST dataset. The right pane shows results from TRE, the state-of-the-art DRE method, which fails to capture the distribution of CIFAR-10. CDM is the first DRE-based method that can successfully learn the distribution of images.

evaluations (NFEs) to calculate the likelihood-ELBO [19], or to approximate the exact likelihood using an ODE solver [43].

DDMs are based on minimum-MSE (MMSE) *denoising*, while DRE methods hinge on optimal *classification*. In this work, we develop a connection between the optimal classifier for predicting the level of white Gaussian noise added to a data sample, and the MMSE denoiser for cleaning such noise. Specifically, we show that the latter can be obtained from the gradient of the former. Utilizing this connection, we propose *classification diffusion model* (CDM), a generative method that combines the formalism of DDMs, but instead of a denoiser, employs a noise-level classifier. CDM is the first instance of a DRE-based method that can successfully generate images beyond MNIST (Fig. 1). In addition, as a DRE method, CDM is inherently capable of outputting the exact log-likelihood in a single NFE. In fact, it achieves state-of-the-art negative-log-likelihood (NLL) results among methods that use a single NFE, and comparable results to computationally-expensive ODE-based methods.

Our experiments shed light on the reasons why DRE methods have failed on complex high-dimensional data to date, and why CDM inherently avoids these challenges. Furthermore, we show that CDM can serve as a more accurate denoiser, in terms of MSE, than a DDM with a similar architecture. This typically translates into better FID scores. Representative generated samples are shown in Fig. 1. We hope that our approach will spark new interest in DRE methods and ultimately unlock their full potential.

## 2  Background

### 2.1  Density Ratio Estimation

Learning data distributions via DRE was first proposed by Gutmann and Hyvärinen [15]. Their *noise contrastive estimation* (NCE) method uses the fact that the ratio between an unknown distribution $p_d(\boldsymbol{x})$ and a known reference distribution $p_n(\boldsymbol{x})$ can be extracted from the optimal binary classifier for discriminating samples from $p_d(\boldsymbol{x})$ and $p_n(\boldsymbol{x})$. Once this ratio is extracted from the classifier, it can be multiplied by the known $p_n(\boldsymbol{x})$ to obtain $p_d(\boldsymbol{x})$. Specifically, let $C$ denote the class of a sample $\boldsymbol{x}$, with $C = 1, 0$ corresponding to the event that $\boldsymbol{x}$ is a sample from $p_d(\boldsymbol{x}), p_n(\boldsymbol{x})$, respectively. The optimal classifier for predicting $C$ from $\boldsymbol{x}$ outputs both $\mathbb{P}(C = 1|\boldsymbol{x})$ and $\mathbb{P}(C = 0|\boldsymbol{x})$. Using Bayes' rule, we can use these values to compute the density ratio

$$\frac{p_d(\boldsymbol{x})}{p_n(\boldsymbol{x})} = \frac{\mathbb{P}(C = 1|\boldsymbol{x})}{\mathbb{P}(C = 0|\boldsymbol{x})}, \tag{1}$$

where we assumed that the classes are balanced, so that $\mathbb{P}(C = 1) = \mathbb{P}(C = 0) = \frac{1}{2}$.

Unfortunately, this method fails in practice when $p_d(\boldsymbol{x})$ and $p_n(\boldsymbol{x})$ differ significantly from one another, as is the case when $p_d(\boldsymbol{x})$ is the distribution of images and $p_n(\boldsymbol{x})$ corresponds to white Gaussian noise. This is because when training a classifier to discriminate between images and noise, the classifier can achieve very high accuracy even without learning meaningful information about images. When this point is reached, the weights of the classifier practically stop updating. Rhodes et al. [35] referred to this issue as the *density-chasm problem*, and suggested to overcome it by making the classification problem more difficult. To do so, their TRE method uses a sequence of distributions $p_{\mathbf{x}_0}(\boldsymbol{x}), p_{\mathbf{x}_1}(\boldsymbol{x}), \ldots, p_{\mathbf{x}_m}(\boldsymbol{x})$, which are closer to one another, such that $p_{\mathbf{x}_m}(\boldsymbol{x})$ is the reference distribution and $p_{\mathbf{x}_0}(\boldsymbol{x})$ is the target distribution. The intermediate distributions $\{p_{\mathbf{x}_i}(\boldsymbol{x})\}_{i=1}^{m-1}$ do not have to be known; the only requirement is that it would be possible to sample from them. For example, $\mathbf{x}_i$ can be defined as $\mathbf{x}_i = \sqrt{\bar{\alpha}_i}\mathbf{x}_0 + \sqrt{1-\bar{\alpha}_i}\mathbf{x}_m$, where $\mathbf{x}_0 \sim p_{\mathbf{x}_0}$, $\mathbf{x}_m \sim p_{\mathbf{x}_m}$, and $\bar{\alpha}_i$ is a sequence that decreases from 1 to 0. Then, using (1), each ratio $p_{\mathbf{x}_i}(\boldsymbol{x})/p_{\mathbf{x}_{i+1}}(\boldsymbol{x})$ can be extracted by training a binary classifier to distinguish between samples from $p_{\mathbf{x}_i}(\boldsymbol{x})$ and $p_{\mathbf{x}_{i+1}}(\boldsymbol{x})$, and the ratio between the target and the reference distributions can be calculated as

$$\frac{p_{\mathbf{x}_0}(\boldsymbol{x})}{p_{\mathbf{x}_m}(\boldsymbol{x})} = \frac{p_{\mathbf{x}_0}(\boldsymbol{x})}{p_{\mathbf{x}_1}(\boldsymbol{x})} \cdot \frac{p_{\mathbf{x}_1}(\boldsymbol{x})}{p_{\mathbf{x}_2}(\boldsymbol{x})} \cdots \frac{p_{\mathbf{x}_{m-2}}(\boldsymbol{x})}{p_{\mathbf{x}_{m-1}}(\boldsymbol{x})} \cdot \frac{p_{\mathbf{x}_{m-1}}(\boldsymbol{x})}{p_{\mathbf{x}_m}(\boldsymbol{x})}. \tag{2}$$

While this method overcomes the *density-chasm-problem* for each pair of consecutive distributions, it still fails in learning the distribution of datasets that are more complicated than MNIST, as illustrated in Fig. 1. This is because each ratio $p_{\mathbf{x}_i}(\boldsymbol{x})/p_{\mathbf{x}_{i+1}}(\boldsymbol{x})$ is extracted from a binary classifier trained only on inputs $\boldsymbol{x}$ from the distributions $p_{\mathbf{x}_i}$ and $p_{\mathbf{x}_{i+1}}$. For instance, the classifier producing the ratio $p_{\mathbf{x}_0}(\boldsymbol{x})/p_{\mathbf{x}_1}(\boldsymbol{x})$ is trained on inputs close to the real data, while the one producing $p_{\mathbf{x}_{m-1}}(\boldsymbol{x})/p_{\mathbf{x}_m}(\boldsymbol{x})$ is trained on inputs close to the reference distribution. This can lead to a mismatch between training and test time, since at inference, all the ratios are evaluated at the same input $\boldsymbol{x}$. Moreover, even if each individual ratio is nearly accurate, the accumulation of small errors can result in a significant overall error. Our method is also based on a classification objective, however it avoids these problems by employing an additional loss, which is based on our main result (Theorem 3.1).

## 2.2 Denoising Diffusion Models

DDMs [38, 19], are a class of generative models that sample from a learned target distribution by gradually denoising white Gaussian noise. More formally, DDMs generate samples by attempting to reverse a forward diffusion process with $T$ steps that starts from a data point $\mathbf{x}_0$ and evolves as $\mathbf{x}_t = \sqrt{\alpha_t}\mathbf{x}_{t-1} + \sqrt{1-\alpha_t}\tilde{\varepsilon}_t$, $t = 1, \ldots, T$, where $\{\tilde{\varepsilon}_t\}$ are iid standard Gaussian vectors. Samples along this forward diffusion process can be equivalently expressed as

$$\mathbf{x}_t = \sqrt{\bar{\alpha}_t}\mathbf{x}_0 + \sqrt{1-\bar{\alpha}_t}\varepsilon_t, \quad \varepsilon_t \sim \mathcal{N}(0, \mathbf{I}), \tag{3}$$

where $\bar{\alpha}_t = \prod_{s=1}^t \alpha_s$. The coefficients $\{\alpha_t\}$ are taken to be such that $\{\bar{\alpha}_t\}$ is a monotonic sequence with $\bar{\alpha}_T \approx 1$. This enforces the density $p_{\mathbf{x}_T}$ to be close to the normal distribution $\mathcal{N}(0, \mathbf{I})$.

The reverse diffusion process is learned by modeling the distributions of $\mathbf{x}_{t-1}$ given $\mathbf{x}_t$ as a Gaussian with mean

$$\mathbb{E}[\mathbf{x}_{t-1}|\mathbf{x}_t] = \frac{1}{\sqrt{\alpha_t}}\left(\mathbf{x}_t - \frac{1-\alpha_t}{\sqrt{1-\bar{\alpha}_t}}\varepsilon_\theta(\mathbf{x}_t, t)\right) \tag{4}$$

and covariance $\sigma_t\mathbf{I}$, where $\varepsilon_\theta(\cdot, \cdot)$ is a neural network and $\{\sigma_t\}$ are fixed hyperparameters. Training is done by minimizing the ELBO loss, which reduces to a series of MSE terms,

$$\mathcal{L}(\theta) = \sum_{t=1}^T \mathbb{E}_{\mathbf{x}_0, \varepsilon_t}\left[\|\varepsilon_\theta(\mathbf{x}_t, t) - \varepsilon_t\|_2^2\right]. \tag{5}$$

At convergence to the optimal solution, the neural network approximates the timestep-dependent posterior mean

$$\varepsilon_\theta(\boldsymbol{x}_t, t) = \mathbb{E}[\varepsilon_t|\mathbf{x}_t = \boldsymbol{x}_t]. \tag{6}$$

To generate samples, DDMs sample $\mathbf{x}_T \sim \mathcal{N}(0, \mathbf{I})$ and then iteratively follow the learned reverse probabilities, terminating with a sample of $\mathbf{x}_0$. In more detail, at each timestep $t$, the model accepts $\mathbf{x}_t$ and outputs a prediction of the noise $\varepsilon_t$ (equivalently, a prediction of the clean signal $\mathbf{x}_0$), from which $\mathbf{x}_{t-1}$ is obtained by sampling from the reverse distribution. The process described above is that underlying the DDPM method [19]. Here we also experiment with DDIM [39] and DPM-Solver [32], which follow a similar structure.

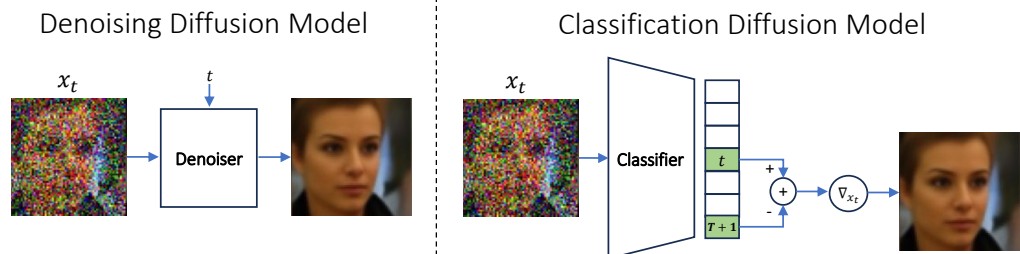

Figure 2: **A diagram of CDM (right) compared with DDM (left).** A DDM functions as an MMSE denoiser conditioned on the noise level, whereas a CDM operates as a classifier. Given a noisy image, a CDM outputs a probability vector predicting the noise level, such that the $t$-th element in this vector is the probability that the noise level of the input image corresponds to timestep $t$ in the diffusion process. A CDM can be used to output the MMSE denoised image by computing the gradient of its output probability vector w.r.t the input image, as we show in Theorem 3.1.

## 3   Method

We start by deriving a relation between classification and denoising, and then use it as the basis for our CDM method. A summary of the notations we use can be found in App. A.

Let the random vector $\mathbf{x}_t$ be defined as in (3) for timesteps $t \in \{1, \dots, T\}$ and set two additional timesteps, $0$ and $T+1$, corresponding to clean images and pure Gaussian noise, respectively. Namely, we define $\bar{\alpha}_0 = 1$ and $\bar{\alpha}_{T+1} = 0$. We denote the density of each $\mathbf{x}_t$ by $p_{\mathbf{x}_t}(\boldsymbol{x})$. Our approach is based on training a classifier that takes as input a noisy sample $\mathbf{x}_t$ and predicts its timestep $t$. Formally, let t be a discrete random variable taking values in $\{0, 1, \dots, T+1\}$, with probability mass function $p_t(t) = \mathbb{P}(t = t)$, and let the random vector $\tilde{\mathbf{x}}$ be the diffusion signal at a random timestep t, namely[1] $\tilde{\mathbf{x}} = \mathbf{x}_t$. Note that the density of each $\mathbf{x}_t$ can be written as $p_{\mathbf{x}_t}(\boldsymbol{x}) = p_{\tilde{\mathbf{x}}|t}(\boldsymbol{x}|t)$ and by the law of total probability, the density of $\tilde{\mathbf{x}}$ is equal to

$$p_{\tilde{\mathbf{x}}}(\boldsymbol{x}) = \sum_{t=0}^{T+1} p_{\mathbf{x}_t}(\boldsymbol{x})\, p_t(t). \tag{7}$$

We are interested in a classifier for predicting t from $\tilde{\mathbf{x}}$. It is well known that given any sample $\boldsymbol{x}$ drawn from (7), the optimal such classifier (in terms of the cross-entropy loss) outputs the probability vector $(p_{t|\tilde{\mathbf{x}}}(0|\boldsymbol{x}), p_{t|\tilde{\mathbf{x}}}(1|\boldsymbol{x}), \dots, p_{t|\tilde{\mathbf{x}}}(T+1|\boldsymbol{x}))$, where $p_{t|\tilde{\mathbf{x}}}(t|\boldsymbol{x}) = \mathbb{P}(t = t|\tilde{\mathbf{x}} = \boldsymbol{x})$. As we now show, the denoiser in (6) corresponds to the gradient of this classifier.

**Theorem 3.1.** *Let $F(\boldsymbol{x}, t) = \log(p_{t|\tilde{\mathbf{x}}}(T+1|\boldsymbol{x})) - \log(p_{t|\tilde{\mathbf{x}}}(t|\boldsymbol{x}))$ with t, $\tilde{\mathbf{x}}$ and $\mathbf{x}_t$ as defined above. Then*

$$\mathbb{E}[\varepsilon_t|\mathbf{x}_t = \boldsymbol{x}_t] = \sqrt{1 - \bar{\alpha}_t}(\nabla_{\boldsymbol{x}_t} F(\boldsymbol{x}_t, t) + \boldsymbol{x}_t) \tag{8}$$

*regardless of the choice of the probability mass function $p_t$, provided that $p_t(t) > 0$ for all t.*

The proof, provided in App. B.1, consists of three key steps:

- Using Bayes rule, we write $p_{\mathbf{x}_t}(\boldsymbol{x})$ as a function of $p_{\mathbf{x}_{T+1}}(\boldsymbol{x})$ and the optimal classifier.
- Then, we take the derivative of the log of both sides and use the fact that $\nabla_{\boldsymbol{x}} \log p_{\mathbf{x}_{T+1}}(\boldsymbol{x})$ has a closed form solution.
- Finally, we use Tweedie's formula [34, 45, 11] to connect between $\nabla_{\boldsymbol{x}} \log p_{\mathbf{x}_t}(\boldsymbol{x})$ and $\mathbb{E}[\varepsilon_t|\mathbf{x}_t = \boldsymbol{x}]$.

Theorem 3.1 suggests that we may train a classifier and use its gradient as a denoiser according to relation (8). This paradigm is illustrated in Fig. 2. Once we have constructed a denoiser, we can apply any desired sampling method (e.g. DDPM, DDIM, etc.) to generate images from the learned distribution. However, as we show in Sec. 4.1, naively training such a classifier with the standard

---

[1]Note the distinction between the notations $\mathbf{x}_t$ and $\mathbf{x}_t$. The former has a random noise level t, while the latter has a fixed noise level $t$.

**Algorithm 1** CDM Training

**Require:** Dataset of training samples $\mathcal{D}$
1: **repeat**
2:     $\boldsymbol{x}_0 \sim \mathcal{D}, t \sim U\{0, \ldots, T{+}1\}, \varepsilon \sim \mathcal{N}(0, \mathbf{I})$
3:     $\boldsymbol{x}_t = \sqrt{\bar{\alpha}_t}\boldsymbol{x}_0 + \sqrt{1 - \bar{\alpha}_t}\varepsilon$
4:     $F_\theta(\boldsymbol{x}_t, t) = f_\theta(\boldsymbol{x}_t)[T+1] - f_\theta(\boldsymbol{x}_t)[t]$
5:     $\varepsilon_\theta(\boldsymbol{x}_t, t) = \sqrt{1 - \bar{\alpha}_t}\left(\nabla_{\boldsymbol{x}_t} F_\theta(\boldsymbol{x}_t, t) + \boldsymbol{x}_t\right)$
6:     take gradient step on
7:     $w_{ce}\mathcal{L}_{\text{CE}}(t, f_\theta(\boldsymbol{x}_t)) + \mathcal{L}_{\text{MSE}}(\varepsilon, \varepsilon_\theta(\boldsymbol{x}_t, t))$
8: **until** converged

**Algorithm 2** DDPM Sampling Using CDM

**Require:** Noise level classifier $f_\theta(\cdot)$
1: sample $\boldsymbol{x}_T \sim \mathcal{N}(0, \mathbf{I})$
2: **for** $t = T, \ldots, 1$ **do**
3:     $F_\theta(\boldsymbol{x}_t, t) = f_\theta(\boldsymbol{x}_t)[T+1] - f_\theta(\boldsymbol{x}_t)[t]$
4:     $\varepsilon_\theta(\boldsymbol{x}_t, t) = \sqrt{1 - \bar{\alpha}_t}\left(\nabla_{\boldsymbol{x}_t} F_\theta(\boldsymbol{x}_t, t) + \boldsymbol{x}_t\right)$
5:     if $t > 1$ then $z \sim \mathcal{N}(0, \mathbf{I})$, else $z = 0$
6:     $\boldsymbol{x}_{t-1} = \frac{1}{\sqrt{\alpha_t}}(\boldsymbol{x}_t - \frac{1-\alpha_t}{\sqrt{1-\bar{\alpha}_t}}\varepsilon_\theta(\boldsymbol{x}_t, t)) + \sigma_t z$
7: **end for**
8: **return** $\boldsymbol{x}_0$

cross-entropy (CE) loss leads to poor results. This is because a classifier may reach a low CE loss even without learning the correct probability $p_{t|\tilde{\mathbf{x}}}(t|\boldsymbol{x})$ for any $t$. This phenomenon can be observed in Fig. 3, which illustrates the reason that existing DRE methods fail to capture the distribution of high-dimensional complex data like images. We discuss this in more detail in Sec. 4.1.

To obtain the correct probability $p_{t|\tilde{\mathbf{x}}}(t|\boldsymbol{x})$ for any $t$, we suggest training the classifier with a combination of a CE loss on its outputs, and an MSE loss on its gradient, according to relation (8). The network's gradient can be efficiently computed using automatic-differentiation. Following Ho et al. [19], we use the same weight for all timesteps in the MSE loss. Our full training scheme is described in Algorithm 1. Here, $f_\theta(\boldsymbol{x})[t]$ denotes the model's $t$-th logit, which serves as an approximation for $\log p_{t|\tilde{\mathbf{x}}}(t|\boldsymbol{x})$ (up to an additive constant that cancels out in the SoftMax operation), and $F_\theta(\boldsymbol{x}, t) = f_\theta(\boldsymbol{x})[T+1] - f_\theta(\boldsymbol{x})[t]$. The added timesteps, corresponding to entries $0$ and $T+1$ of the classifier, are trained only using the CE loss. This is because the prediction of the noise is trivial when $\text{t} = T + 1$ and meaningless when $\text{t} = 0$ (since there is no noise). Importantly, this behavior is automatically achieved without any modification to the algorithm. Specifically, in line 4 of the algorithm, $F_\theta(\boldsymbol{x}_{T+1}, T + 1) = 0$, and in line 5, $\varepsilon_\theta(\boldsymbol{x}_0, 0) = 0$, preventing the MSE loss from updating the weights for these timesteps.

Algorithm 2 shows how to generate samples with CDM using the DDPM sampler (a similar approach can be used with other samplers). Note that each step $t$ in DDPM sampling using CDM is given by

$$\boldsymbol{x}_{t-1} = \sqrt{\alpha_t}\boldsymbol{x}_t - \frac{1 - \alpha_t}{\sqrt{\alpha_t}}\nabla_{\boldsymbol{x}_t} F_\theta(\boldsymbol{x}_t, t) + \sigma_t z, \tag{9}$$

where $z \sim \mathcal{N}(0, \mathbf{I})$. Therefore, each step steers the process in the direction that maximizes the probability of noise level $t$, while minimizing the probability of noise level $T + 1$. This can be thought of as taking a gradient step with size $(1 - \alpha_t)/\sqrt{\alpha_t}$, followed by a step in a random exploration direction with magnitude $\sigma_t$, similarly to Langevin dynamics.

### 3.1 Exact Likelihood Calculation in a Single Step

To compute the likelihood of a given sample, DDMs are required to perform many NFEs in order to compute a lower bound on the log likelihood using the ELBO [19], or can approximate the exact likelihood [43] using an ODE solver based on repeated evaluations of the network. In contrast, as a DRE-based method, a CDM is able to calculate the exact likelihood in a single NFE. In fact, a CDM can compute the likelihood w.r.t. the distribution $p_{\mathbf{x}_t}$ of noisy images, for any desired timestep $t$. Specifically, we have the following (see proof in App. B.3)

**Theorem 3.2.** *For any $t \in \{0, 1, \ldots, T + 1\}$,*

$$p_{\mathbf{x}_t}(\boldsymbol{x}) = \frac{p_t(T+1)}{p_t(t)} \frac{p_{t|\tilde{\mathbf{x}}}(t|\boldsymbol{x})}{p_{t|\tilde{\mathbf{x}}}(T+1|\boldsymbol{x})} \mathcal{N}(\boldsymbol{x}; 0, \mathbf{I}), \tag{10}$$

*where $\mathcal{N}(\cdot; 0, \mathbf{I})$ is the probability density function of a standard multivariate Gaussian distribution.*

Note that the first term in (10) only depends on the pre-selected probability mass function $p_t$ (which we choose to be uniform in our experiments), and the second term can be obtained from the $t$-th and $(T+1)$-th entries of the vector at the output of the classifier (after applying SoftMax). This implies

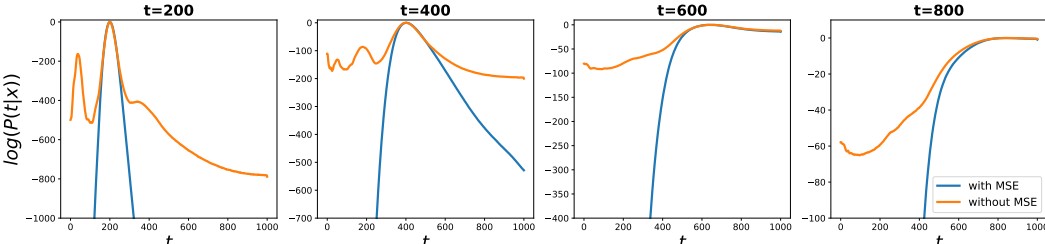

Figure 3: **Comparison between the log probability of a noise-level classifier trained using the CE loss alone and a model trained using CE and MSE**. Since the SoftMax operator is invariant to an additive factor, we subtract the maximal value from the vector (i.e., $f_\theta(\boldsymbol{x}_t) \leftarrow f_\theta(\boldsymbol{x}_t) - \max(f_\theta(\boldsymbol{x}_t))$) for visualization. We utilize the connection we developed between the optimal classifier and the MMSE denoiser to incorporate the MSE loss in DRE training, as depicted in Algorithm 1. As evident, without considering MSE, the prediction accuracy of the classifier is limited to the vicinity of the correct label, unlike the model trained using both CE and MSE, which yields accurate predictions globally. The essence of an optimal classifier lies in its capability to predict the correct probability vector for all entries, rather than solely for the correct label. As can be seen, this necessitates the incorporation of the MSE loss.

that we can calculate the likelihood of any given image $\boldsymbol{x}$ w.r.t. to the density $p_{\mathbf{x}_t}$ of noisy images for any noise level $t$. In particular, $p_{\mathbf{x}_0}$ is the density of clean images. Thus, by choosing $t = 0$ we can calculate the likelihood of clean images. In App. D we present a simple validation of our efficient likelihood computation by applying it to toy examples with known densities, allowing us to compute the analytical likelihood and verify that our computations align with theoretical expectations.

# 4 Experiments

We train several CDMs on two common datasets. For CIFAR-10 [26] we train both a class conditional model and an unconditional model. We also train a similar model for CelebA [31], using face images of size $64 \times 64$. In Sec. 4.1, we demonstrate why existing DRE methods fail on complex high-dimensional data like images, and show how the incorporation of the MSE loss in our method, according to Theorem 3.1, overcomes these challenges. In Sec. 4.2, we compare our method with pre-trained DDMs with similar architectures, to disentangle the benefits of our method from other variables. We evaluate the performance of CDM as a denoiser, assess its generation quality using FID [17], and measure its likelihood modeling capabilities using NLL. Finally, in Sec. 4.3, we demonstrate the use of different noise schedulers, one specifically tuned for likelihood estimation, and one corresponding to the flow matching optimal-transport scheme [29, 30]. We show that incorporating these schedulers into our method leads to state-of-the-art NLL results among methods capable of outputting the likelihood in a single forward pass.

## 4.1 The Importance of Using Both Losses for Achieving an Optimal Classifier

In Theorem 3.1, we established that the MMSE denoiser corresponds to the gradient of the optimal noise-level classifier. A natural question is whether we can train our model only with the MSE loss. Unfortunately, the answer is negative. This is because the MSE achieved by the model does not change if we add a function of $t$ to its output, as such an additive term vanishes when taking the gradient with respect to $\boldsymbol{x}$. The CE loss is important for removing this degree of freedom. Namely, without the CE loss, the model can function as a denoiser but is useless for the purpose of outputting the likelihood in a single step.

Can we train the model only with the CE loss, then? In theory, training the model only with the CE loss should be sufficient. However, as we will demonstrate next, incorporating MSE is crucial in practice for achieving an optimal classifier.

Table 1 reports the MSE, CE and classification accuracy achieved by models trained with different losses. We emphasize that the model trained using only MSE in this comparison, is a CDM model trained using Algorithm 1, and is not equivalent to a DDM model trained using (5). As evident from

Table 1: **The importance of using both losses in CDM.** We demonstrate the importance of using both the CE and MSE losses at training. We report the results for CIFAR-10 test-set. FID is reported on 50k samples which were generated using DDIM scheduler with 50 steps. As shown by Rhodes et al. [35], to avoid the *density-chasm problem*, the classification problem should be sufficiently hard to avoid trivial classifier solutions. This leads to low classification accuracy results.

| Training Loss | Classification Acc ↑ | CE ↓ | MSE ↓ | FID ↓ | NLL ↓ |
|---|---|---|---|---|---|
| CE | 6.97% | 4.49 | 0.225 | 329 | 8.27 |
| MSE | 0% | 1659 | **0.028** | 7.65 | 9.32 |
| Both | **8.34**% | **4.37** | **0.028** | **7.56** | **3.38** |

the table, when using only the CE loss, the MSE is high, and when using only the MSE loss the CE and classification accuracy are poor. An important point to notice is that even when training with the CE loss, the classifier's accuracy is rather low (though greater than the 0% achieved when training only with the MSE loss). This is a key prerequisite for making DRE methods work. Specifically, as shown in [35], the classification problem should be sufficiently hard in order to avoid the *density-chasm problem*, otherwise the classifier can easily discriminate between the classes even without having learned the correct density ratio. Yet, as we illustrate next, only making the classification problem harder is still insufficient for learning the probability $p_{\mathrm{t}|\tilde{\mathbf{x}}}(t|\boldsymbol{x})$ with only the CE loss.

Figure 3 shows the logits $f_\theta(\boldsymbol{x}_t)$ for noisy images with different noise levels, comparing a model trained using CE to a model trained with both CE and MSE. As can be seen, in both scenarios the prediction near the true label is the same, namely the CE works well in the vicinity of the correct noise level. However, the model trained without the MSE loss exhibits significantly higher predicted logits for more distant noise levels compared to the model trained using both CE and MSE. Moreover, the logits of the model trained without MSE do not decrease monotonically as the distance from the actual noise level increases, which is in contrast with the expected behavior. This demonstrates the importance of the MSE loss for obtaining good prediction globally. As can be seen in Theorem 3.1, the denoiser at timestep $t$ depends on the predictions of the classifier in both the $t$-th and the $(T+1)$-th entries. Therefore, the addition of the MSE loss enforces the classifier to achieve accurate predictions in both entries, thereby ensuring accurate predictions globally.

## 4.2 Denoising Results, Image Quality and Negative Log Likelihood

We compare our method with pre-trained DDMs of similar architectures. Since CDM is a classifier and DDM is a timestep-conditional denoiser, we take the architecture of our CDM to be identical to the DDM, except for altering the last two layers to output a vector of logits, and removing all timestep conditioning layers. These changes have a negligible effect on the number of parameters in the model. For more details please refer to App. C.

As shown in Fig. 4, the denoising performance of our CDM surpasses that of pre-trained DDMs at high noise levels, and is comparable to them at lower noise levels. These quantitative results are corroborated by the qualitative examples in Fig. 5, which showcase image denoising results across various noise levels.

The good denoising performance of CDM translates into high quality image generation. This is illustrated qualitatively in Fig. 1, which shows samples from models trained on CelebA and on CIFAR-10 (unconditional). To quantitatively compare the generation quality of CDM to that of pre-trained DDMs, we use 50k FID [17] against the train-set. For both CDMs and DDMs, we compare images sampled using the DDPM [19], DDIM [39], and DPM-Solver (DPMS) [32] samplers, using 1000, 50, and 25 timesteps, respectively. The results, shown in Table 2, demonstrate that CDM is at least comparable to pre-trained DDMs in image quality, outperforming them in most cases.

Additionally, we evaluate CDM's effectiveness in applying classifier-free guidance (CFG) [18] for conditional sampling tasks. As expected, incorporating CFG improves image quality beyond unconditional generation, as reflected in the conditional CIFAR-10 FID results of Table 2. More details and qualitative results are provided in Appendix C.5. These results showcase the effectiveness of CDM for image generation, showing it to be equal or better than a similar DDM.

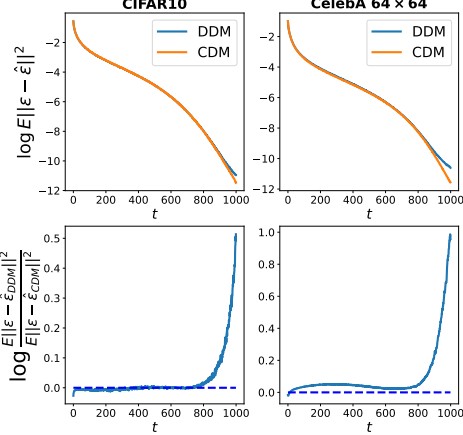

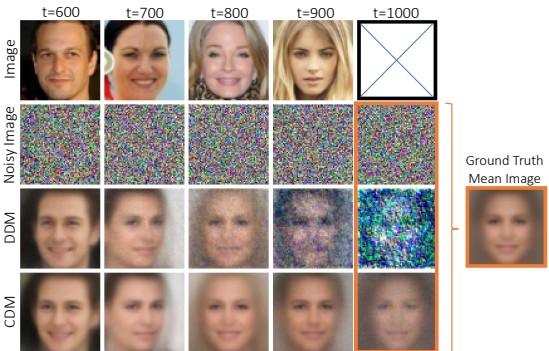

Figure 4: **Denoising performance.** The plots show the MSEs (top) and the ratio between the MSEs (bottom) achieved by a pre-trained DDM and by a CDM with the same architecture, as a function of the noise level (timestep $t$). The CDM significantly outperforms the pre-trained DDM at high noise levels, while demonstrating comparable performance at lower noise levels.

Figure 5: **Denoising results.** The figure depicts a comparison between denoising results on the CelebA dataset for several different noise levels, obtained with a CDM and with a pre-trained DDM with the same architecture. The right column shows the models' predictions for pure Gaussian noise, which should theoretically be the expectation of the prior distribution. As observed, DDM outputs a highly noisy image, whereas CDM generates an image much closer to the mean of the dataset.

Finally, we calculate log-likelihoods and compare our NLL results to recent leading methods in Table 3. CDM demonstrates comparable performance on NLL estimation for CIFAR-10 compared to DDMs. Notably, CDM stands out as a more efficient method than existing ones, requiring only a single forward pass for NLL computation. Table 3 also includes CDM(unif.), and CDM(OT) on which we elaborate in Sec. 4.3 below. These variants of our model, improve the NLL predictions and achieve state-of-the-art results among methods requiring a single step.

### 4.3 Different Noise Scheduling for Better Likelihood Estimation

To see the effect of using a timestep scheduler tuned for DRE tasks, we repeat the CIFAR-10 unconditional experiment, with a different noise scheduler. Following Rhodes et al. [35] we use the scheduler defined as $\sqrt{1 - \bar{\alpha}_t} = \frac{t}{T+1}$ and choose $T = 1000$ similarly to our previous experiments. Utilizing this scheduler, we achieve a better NLL of 2.98 at the expense of a higher FID of 10.28, when using the DDIM sampler with 50 steps. This trade-off highlights that the scheduler optimal for learning the data distribution may not be ideal for sampling.

To further explore the influence of the noise scheduler, we train and evaluate a CDM with the flow-matching optimal-transport (OT) scheduler [29, 30] on unconditional CIFAR-10. In this scheduler, $\mathbf{x}_t = \frac{T-t}{T}\mathbf{x}_0 + \frac{t}{T}\varepsilon_t$, where $\varepsilon_t \sim \mathcal{N}(0, \mathbf{I})$ and $t \in \{0, \dots, T\}$. This scheduler leads to a state-of-the-art single-step NLL of 2.89 and to an FID of 7.07 with 1000 sampling steps. Please see App. B.4 for more details.

Future research could explore schedulers aimed at further enhancing the NLL. Further analysis of the difference between the schedulers from a classification perspective can be found in App. E.2

## 5 Related Work

Using the concept of DRE for learning data distributions was initially studied by Gutmann and Hyvärinen [15]. Their noise contrastive estimation (NCE) method approximates the ratio between the density of the data distribution and that of white Gaussian noise. However, it struggles in practical scenarios where the gap between these distributions is large, as is the case for natural images [35]. Conditional noise contrastive estimation (CNCE) [4] is a slightly improved version of NCE, in which

Table 2: **Image generation quality.** We compare the FID (lower is better) achieved by a DDM and a CDM using three sampling schemes for CelebA and CIFAR-10. For conditional CIFAR-10 we train a DDM ourselves, as no model in the original implementation [19] supports CFG.

| Sampling Method | Model | |
|---|---|---|
| | DDM | CDM |
| CelebA $64 \times 64$ | DDM | CDM |
| DDIM Sampler, 50 steps | 8.47 | **4.78** |
| DDPM Sampler, 1000 steps | 4.13 | **2.51** |
| 2nd order DPMS, 25 steps | 6.16 | **4.45** |
| Uncond CIFAR-10 | DDM | CDM |
| DDIM Sampler, 50 steps | **7.19** | 7.56 |
| DDPM Sampler, 1000 steps | 4.77 | **4.74** |
| 2nd order DPMS, 25 steps | **6.91** | 7.29 |
| Cond CIFAR-10 | DDM | CDM |
| DDIM Sampler, 50 steps | 5.92 | **5.08** |
| DDPM Sampler, 1000 steps | 4.70 | **3.66** |
| 2nd order DPMS, 25 steps | 5.87 | **4.87** |

Table 3: **NLL (bits/dim) calculated on the CIFAR-10 test-set.** For each model we specify the number of NFEs required for calculating the NLL. CDM achieves state-of-the-art NLL among methods that use a single NFE.

| Model | NLL $\downarrow$ | NFE |
|---|---|---|
| iResNet [2] | 3.45 | 100 |
| FFJORD [13] | 3.40 | $\sim$3K |
| MintNet [41] | 3.32 | 120 |
| FlowMatching [29] | 2.99 | 142 |
| VDM [22] | **2.65** | 10K |
| DDPM ($L$) [19] | $\leq$3.70 | 1K |
| DDPM ($L_{simple}$) [19] | $\leq$3.75 | 1K |
| DDPM (SDE) [43] | 3.28 | $\sim$200 |
| DDPM++ cont. [43] | 2.99 | $\sim$200 |
| RealNVP [10] | 3.49 | 1 |
| Glow [24] | 3.35 | 1 |
| Residual Flow [6] | 3.28 | 1 |
| CDM | 3.38 | 1 |
| CDM(unif.) | **2.98** | 1 |
| CDM(OT) | **2.89** | 1 |

the classification problem is designed to be harder. Specifically, CNCE is based on training a classifier to predict the order of a pair of samples with closer densities, e.g. achieved by pairing a data sample with its noisy version.

Telescoping density-ratio estimation (TRE), proposed by Rhodes et al. [35], avoids direct classification between data and noise. Instead, it uses a gradual transition between those two distributions, and trains a classifier to distinguish between samples from every pair of adjacent densities. Such a classifier learns the ratio between adjacent distributions, and the overall ratio between the data and noise distributions is computed by multiplying all intermediate ratios.

Choi et al. [7] extended this idea from a finite set of intermediate densities to an infinite continuum. This was accomplished by deriving a link between the density ratios for infinitesimally close distributions and the principles of score matching [23, 40, 42], motivating the training of a model to predict the time score $\frac{\partial}{\partial t} \log p_{\mathbf{x}_t}$. In contrast, we draw a different connection which shows that an MMSE denoiser can be obtained as the gradient of an optimal noise level classifier. Also, to obtain the log ratio between the target and reference distributions, Choi et al. [7] need to solve an integral over the time-score using an ODE solver, while in our method this ratio can be calculated in a single NFE.

Yair and Michaeli [47] extended the concept of TRE, proposing the training of a single noise level classifier instead of training a binary classifier for each pair of neighboring densities. While this method is conceptually similar to ours, our approach distinguishes itself by incorporating the MSE loss as outlined in Theorem 3.1. As demonstrated in our experiments, this proves to be crucial for achieving an optimal classifier and high-quality image generation.

# 6 Discussion and Conclusion

We developed an analytical connection between an MSE-optimal denoiser for removing white Gaussian noise and a cross-entropy-optimal classifier for predicting the noise level. We used this connection to propose CDM – a DRE based generative technique that is based on a noise-level classifier. Importantly, our classifier is trained using both a classification loss (CE) and a regression loss (MSE). We showed that this key component is what sets CDM apart from existing DRE based methods, and makes it the first instance of a DRE-based technique that can successfully generate images beyond MNIST.

Our approach is not free of limitations. A key challenge is that CDMs can be more computationally expensive than DDMs. Indeed, while DDMs require a single forward pass for each denoising step, CDMs require both a forward pass and a backward pass. Nevertheless, the computational cost of performing a forward and a backward pass through a network depends on its architecture. In this work, we chose to use the same architecture as that used by DDPM [19], in order to isolate the impact of our algorithmic approach from the choice of the model architecture when comparing to DDMs. However, an important future direction would be to explore architectures that are particularly optimized for CDMs and that alleviate the gap in computational complexity. Such architectures should have the property that performing a forward pass and a backward pass through them is computationally similar to performing only a forward pass in a regular DDM. This could potentially be achieved *e.g.*, by relying only on the encoder part of the U-Net. However, we leave this exploration for future work.

**Broader Impact**    CDM is a generative model and thus may potentially suffer from the same limitations as other generative techniques. These include biases in the generated images, as well as malicious and offensive use, such as creating Deepfakes for disinformation. However, CDMs may also impact domains that rely on generative models in a positive way. This is because, different from most generative models, CDMs are able to compute the likelihood for any input in a single step. This may be used *e.g.*, for out-of-distribution detection or for ranking the likelihoods of different restored images in image restoration tasks. Such capabilities may be crucial in fields like medical imaging.

## Acknowledgments

This research was partially supported by the Israel Science Foundation (ISF) under Grant 2318/22.

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

## A  Notation

Table 4: Notation

| Notation | Description | Comments |
|---|---|---|
| t | The random variable over $t \in \{0, 1, \dots, T+1\}$ with distribution $p_t(t)$ | |
| $p_t(t)$ | The distribution of the random variable t | $\mathbb{P}(\mathsf{t} = t)$ |
| $\mathbf{x}_t$ | The diffusion signal at a specific timestep $\mathsf{t} = t$ | |
| $\tilde{\mathbf{x}}, \mathbf{x}_t$ | The diffusion signal at a random timestep t | |
| $p_{\mathbf{x}_t}(\boldsymbol{x})$ | The distribution of noisy images with noise level $\mathsf{t} = t$ | $p_{\tilde{\mathbf{x}}|\mathsf{t}}(\boldsymbol{x}|t)$ |
| $p_{\tilde{\mathbf{x}}}(\boldsymbol{x}), p_{\mathbf{x}_t}(\boldsymbol{x})$ | The joint distribution of noisy images among all the noise levels. | $\sum_{t=0}^{T+1} p_{\mathbf{x}_t}(\boldsymbol{x}) p_t(t)$ |
| $F(\boldsymbol{x}, t)$ | $\log(p_{t|\tilde{\mathbf{x}}}(T+1|\boldsymbol{x})) - \log(p_{t|\tilde{\mathbf{x}}}(t|\boldsymbol{x}))$ | |
| $f_\theta(\boldsymbol{x}_t)$ | Logits vector that is produced by our model | $f_\theta(\boldsymbol{x}_t)[t] \approx \log(p_{t|\tilde{\mathbf{x}}}(t|\boldsymbol{x}))$ |
| $F_\theta(\boldsymbol{x}_t, t)$ | Model approximation of $F(\boldsymbol{x}, t)$ | $f_\theta(\boldsymbol{x}_t)[T+1] - f_\theta(\boldsymbol{x}_t)[t]$ |

## B  Proofs

### B.1  Theorem 3.1 Proof

Using Bayes rule,

$$p_{\mathbf{x}_t}(\boldsymbol{x}) = p_{\tilde{\mathbf{x}}|\mathsf{t}}(\boldsymbol{x}|t) = \frac{p_{t|\tilde{\mathbf{x}}}(t|\boldsymbol{x}) p_{\tilde{\mathbf{x}}}(\boldsymbol{x})}{p_t(t)}. \tag{11}$$

In particular, for $t = T + 1$, this relation reads

$$p_{\mathbf{x}_{T+1}}(\boldsymbol{x}) = \frac{p_{t|\tilde{\mathbf{x}}}(T+1|\boldsymbol{x}) p_{\tilde{\mathbf{x}}}(\boldsymbol{x})}{p_t(T+1)}. \tag{12}$$

Combining (11) and (12) yields

$$p_{\mathbf{x}_t}(\boldsymbol{x}) = \frac{p_t(T+1)}{p_t(t)} \frac{p_{t|\tilde{\mathbf{x}}}(t|\boldsymbol{x})}{p_{t|\tilde{\mathbf{x}}}(T+1|\boldsymbol{x})} p_{\mathbf{x}_{T+1}}(\boldsymbol{x}). \tag{13}$$

Taking the logarithm of both sides, we have

$$\log(p_{\mathbf{x}_t}(\boldsymbol{x})) = \log\left(\frac{p_t(T+1)}{p_t(t)}\right) + \log\left(\frac{p_{t|\tilde{\mathbf{x}}}(t|\boldsymbol{x})}{p_{t|\tilde{\mathbf{x}}}(T+1|\boldsymbol{x})}\right) + \log(p_{\mathbf{x}_{T+1}}(\boldsymbol{x})). \tag{14}$$

Taking the gradient of both sides w.r.t $\boldsymbol{x}$, and noting that the first term on the right hand side does not depend on $\boldsymbol{x}$, we get that

$$\nabla_{\boldsymbol{x}} \log(p_{\mathbf{x}_t}(\boldsymbol{x})) = \nabla_{\boldsymbol{x}} \log(p_{t|\tilde{\mathbf{x}}}(t|\boldsymbol{x})) - \nabla_{\boldsymbol{x}} \log(p_{t|\tilde{\mathbf{x}}}(T+1|\boldsymbol{x})) + \nabla_{\boldsymbol{x}} \log(p_{\mathbf{x}_{T+1}}(\boldsymbol{x})). \tag{15}$$

Since $p_{\mathbf{x}_{T+1}}(\boldsymbol{x}) = \mathcal{N}(\boldsymbol{x}; 0, \mathbf{I})$, we have that $\nabla_{\boldsymbol{x}} \log(p_{\mathbf{x}_{T+1}}(\boldsymbol{x})) = -\boldsymbol{x}$. Therefore, overall we have

$$\nabla_{\boldsymbol{x}} \log(p_{\mathbf{x}_t}(\boldsymbol{x})) = \nabla_{\boldsymbol{x}} \log(p_{t|\tilde{\mathbf{x}}}(t|\boldsymbol{x})) - \nabla_{\boldsymbol{x}} \log(p_{t|\tilde{\mathbf{x}}}(T+1|\boldsymbol{x})) - \boldsymbol{x}. \tag{16}$$

As for the left hand side of (15), since $\mathbf{x}_t = \sqrt{\bar{\alpha}_t} \mathbf{x}_0 + \sqrt{1 - \bar{\alpha}_t} \varepsilon_t$ and $\varepsilon_t \sim \mathcal{N}(0, \mathbf{I})$, using Tweedie's formula [34, 45, 11] it can be shown that

$$\nabla_{\boldsymbol{x}} \log(p_{\mathbf{x}_t}(\boldsymbol{x})) = -\frac{1}{\sqrt{1 - \bar{\alpha}_t}} \mathbb{E}[\varepsilon_t | \mathbf{x}_t = \boldsymbol{x}]. \tag{17}$$

For completeness we provide the full proof for (17) in App. B.2. Substituting (16) into (17) and multiplying both sides by $\sqrt{1 - \bar{\alpha}_t}$ gives

$$\mathbb{E}[\varepsilon_t | \mathbf{x}_t = \boldsymbol{x}] = \sqrt{1 - \bar{\alpha}_t} \left( \nabla_{\boldsymbol{x}} \log(p_{t|\tilde{\mathbf{x}}}(T+1|\boldsymbol{x})) - \nabla_{\boldsymbol{x}} \log(p_{t|\tilde{\mathbf{x}}}(t|\boldsymbol{x})) + \boldsymbol{x} \right), \tag{18}$$

which completes the proof.

Note that the proof is correct for any choice of $p_t$ (provided that $p_t(t) > 0$ for all $t \in \{1, .., T+1\}$), since the term that depends on $p_t$ does not depend on $\boldsymbol{x}$ and thus drops when taking the gradient. In practice, as mentioned in the main text, we chose to use $\mathsf{t} \sim U(\{0, 1, ..., T+1\})$.

## B.2 Proof of Tweedie's Formula

Let us show that if $\mathbf{y} = \mu\mathbf{x} + \sigma\mathbf{z}$ with $\mathbf{z} \sim \mathcal{N}(0, \mathbf{I})$ statistically independent of $\mathbf{x}$, then $\nabla_{\boldsymbol{y}} \log(p_{\mathbf{y}}(\boldsymbol{y})) = -\frac{1}{\sigma}\mathbb{E}[\mathbf{z}|\mathbf{y} = \boldsymbol{y}]$.

From the law of total probability,

$$
\begin{aligned}
p_{\mathbf{y}}(\boldsymbol{y}) &= \int p_{\mathbf{y}|\mathbf{x}}(\boldsymbol{y}|\boldsymbol{x})p_{\mathbf{x}}(\boldsymbol{x})d\boldsymbol{x} \\
&= \int \frac{1}{(2\pi)^{d/2}\sigma^d} \exp\left\{-\frac{1}{2\sigma^2}\|\boldsymbol{y}-\mu\boldsymbol{x}\|^2\right\} p_{\mathbf{x}}(\boldsymbol{x})d\boldsymbol{x}.
\end{aligned}
\tag{19}
$$

Taking the gradient w.r.t. $\boldsymbol{y}$ gives

$$
\begin{aligned}
\nabla_{\boldsymbol{y}} p_{\mathbf{y}}(\boldsymbol{y}) &= \int -\frac{1}{\sigma^2}(\boldsymbol{y}-\mu\boldsymbol{x})\frac{1}{(2\pi)^{d/2}\sigma^d} \exp\left\{-\frac{1}{2\sigma^2}\|\boldsymbol{y}-\mu\boldsymbol{x}\|^2\right\} p_{\mathbf{x}}(\boldsymbol{x})d\boldsymbol{x} \\
&= \int -\frac{1}{\sigma^2}(\boldsymbol{y}-\mu\boldsymbol{x})p_{\mathbf{y}|\mathbf{x}}(\boldsymbol{y}|\boldsymbol{x})p_{\mathbf{x}}(\boldsymbol{x})d\boldsymbol{x} \\
&= \int -\frac{1}{\sigma^2}(\boldsymbol{y}-\mu\boldsymbol{x})p_{\mathbf{x}|\mathbf{y}}(\boldsymbol{x}|\boldsymbol{y})p_{\mathbf{y}}(\boldsymbol{y})d\boldsymbol{x}.
\end{aligned}
\tag{20}
$$

Dividing both sides by $p_{\mathbf{y}}(\boldsymbol{y})$, we get

$$
\begin{aligned}
\nabla_{\boldsymbol{y}} \log p_{\mathbf{y}}(\boldsymbol{y}) &= \int -\frac{1}{\sigma^2}(\boldsymbol{y}-\mu\boldsymbol{x})\, p_{\mathbf{x}|\mathbf{y}}(\boldsymbol{x}|\boldsymbol{y})d\boldsymbol{x} \\
&= -\frac{1}{\sigma^2}\mathbb{E}[\mathbf{y}-\mu\mathbf{x}|\mathbf{y} = \boldsymbol{y}] \\
&= -\frac{1}{\sigma}\mathbb{E}[\mathbf{z}|\mathbf{y} = \boldsymbol{y}].
\end{aligned}
\tag{21}
$$

Now, substituting $\mu = \sqrt{\bar{\alpha}_t}$, $\sigma = \sqrt{1-\bar{\alpha}_t}$, $\mathbf{y} = \mathbf{x}_t$, $\boldsymbol{y} = \boldsymbol{x}_t$, and $\mathbf{z} = \varepsilon_t$, leads to

$$
\nabla_{\boldsymbol{x}_t} \log p_{\mathbf{x}_t}(\boldsymbol{x}_t) = -\frac{1}{\sqrt{1-\bar{\alpha}_t}}\mathbb{E}[\varepsilon_t|\mathbf{x}_t = \boldsymbol{x}_t],
\tag{22}
$$

demonstrating (17).

## B.3 Proof of Theorem 3.2

Substituting $p_{\mathbf{x}_{T+1}}(\boldsymbol{x}) = \mathcal{N}(\boldsymbol{x}; 0, \mathbf{I})$ into (13) leads to

$$
p_{\mathbf{x}_t}(\boldsymbol{x}) = \frac{p_t(T+1)}{p_t(t)} \frac{p_{t|\tilde{\mathbf{x}}}(t|\boldsymbol{x})}{p_{t|\tilde{\mathbf{x}}}(T+1|\boldsymbol{x})} \mathcal{N}(\boldsymbol{x}; 0, \mathbf{I}),
\tag{23}
$$

which proves Theorem 3.2.

## B.4 Proof of CDM for the Flow Matching Optimal Transport Scheduler

In the case of the flow matching optimal-transport scheduler [29, 30], $\mathbf{x}_t$ is defined as

$$
\mathbf{x}_t = \frac{T-t}{T}\mathbf{x}_0 + \frac{t}{T}\varepsilon_t,
\tag{24}
$$

where $\varepsilon_t \sim \mathcal{N}(0, \mathbf{I})$ and $t \in \{0, \dots, T\}$. In this case, the objective of conditional flow matching is [29, 30]

$$
\mathcal{L} = \sum_{t=1}^{T} \mathbb{E}_{\mathbf{x}_0, \varepsilon_t} \left[ v_t(\boldsymbol{x}) - \frac{1}{T}(\varepsilon_t - \mathbf{x}_0) \right]
\tag{25}
$$

and the global minimum of this objective is achieved by

$$v_t(\boldsymbol{x}) = \frac{1}{T} \cdot \mathbb{E}[\varepsilon_t - \mathbf{x}_0 | \mathbf{x}_t = \boldsymbol{x}]. \tag{26}$$

We will start by expressing this solution in terms of the optimal denoiser $\mathbb{E}[\varepsilon_t | \mathbf{x}_t]$. Substituting (24) into (26) and using the fact that $\mathbb{E}[\mathbf{x}_t | \mathbf{x}_t = \boldsymbol{x}] = \boldsymbol{x}$ gives

$$v_t(\boldsymbol{x}) = \frac{1}{T - t} \cdot (\mathbb{E}[\varepsilon_t | \mathbf{x}_t = \boldsymbol{x}] - \boldsymbol{x}). \tag{27}$$

Next, following exactly the same logic as in App. B.1, we can write

$$\mathbb{E}[\varepsilon_t | \mathbf{x}_t = \boldsymbol{x}] = \frac{t}{T} \cdot \left( \nabla_{\boldsymbol{x}} \log(p_{t|\tilde{\mathbf{x}}}(T|\boldsymbol{x})) - \nabla_{\boldsymbol{x}} \log(p_{t|\tilde{\mathbf{x}}}(t|\boldsymbol{x})) + \boldsymbol{x} \right), \tag{28}$$

Finally, by substituting (28) into (27) we get

$$v_t(\boldsymbol{x}) = \frac{t/T}{T(1 - t/T)} \left( \nabla_{\boldsymbol{x}} \log(p_{t|\tilde{\mathbf{x}}}(T|\boldsymbol{x})) - \nabla_{\boldsymbol{x}} \log(p_{t|\tilde{\mathbf{x}}}(t|\boldsymbol{x})) \right) - \frac{1}{T} \cdot \boldsymbol{x}. \tag{29}$$

## C   Implementation details

### C.1   Architectures

For a fair comparison, for each dataset we used the same architecture for our method and for DDMs. As baslines we took the pre-trained model for CelebA $64 \times 64$ from the DDIM Official Github [39] and the EMA pre-trained model for CIFAR-10 from the pytorch diffusion repository, who converted the pre-trained model from the Official DDPM implementation from tensorflow to pytorch.

For conditional CIFAR-10 we trained the DDM model by ourselves because there exist no pre-trained models for CIFAR-10 capable of handling CFG. We used the same architecture from [19] for both models and trained them for the same number of iterations (more details are in App. C.2.2). To condition the model on class labels, we learned an embedding for each class using nn.Embedding and injected it at all points where the time embedding was originally applied. In the case of DDM, we added the class embedding to the time embedding, while for CDM, we replaced the time embedding with the class embedding.

In contrast to DDM architectures, our model does not need the layers that process the timestep input so we removed them. In addition, our model outputs a probability vector in contrast to DDMs which output an image, therefore, we replaced the last convolution layer that reduces the number of channels to 3 in the original architecture by a convolution layers outputs 1024 and 512 channels for CelebA $64 \times 64$ and CIFAR-10, respectively. Following this layer, we performed global average pooling and added a linear layer with an output dimension of $T + 2$. The resulting change in the number of parameters is negligible.

Inspired by Yair and Michaeli [47], we added a non-learned linear transformation at the output of the network, which performs cumulative-summation (cumsum). This enforces (for the optimal classifier) the $t$-th output of the model before this layer to be $\log r_t(\boldsymbol{x}) = \log \frac{p_{t|\tilde{\mathbf{x}}}(t|\boldsymbol{x})}{p_{t|\tilde{\mathbf{x}}}(t-1|\boldsymbol{x})} = \log p_{t|\tilde{\mathbf{x}}}(t|\boldsymbol{x}) - \log p_{t|\tilde{\mathbf{x}}}(t - 1|\boldsymbol{x})$ for $t \neq 0$ and $r_0(\boldsymbol{x}) = \log p_{t|\tilde{\mathbf{x}}}(0|\boldsymbol{x})$, so that after the cumsum layer, the $t$-th output is $\sum_{i=0}^{t} \log r_i(\boldsymbol{x}) = \log p_{t|\tilde{\mathbf{x}}}(t|\boldsymbol{x})$.

### C.2   Hyperparamters

#### C.2.1   CelebA $64 \times 64$

We trained the model for 500k iterations with a learning rate of $1 \cdot 10^{-4}$. We started with a linear warmup of 5k iterations and reduced the learning rate by a factor of 10 after every 200k iterations. The typical value of the CE loss after convergence was $\sim 3.8$ while the MSE loss was $\sim 0.0134$ so we chose to give the CE loss a weight of $0.001$ to ensure the values of both losses have the same order of magnitude. In addition We used EMA with a factor of $0.9999$, as done in the baseline model.

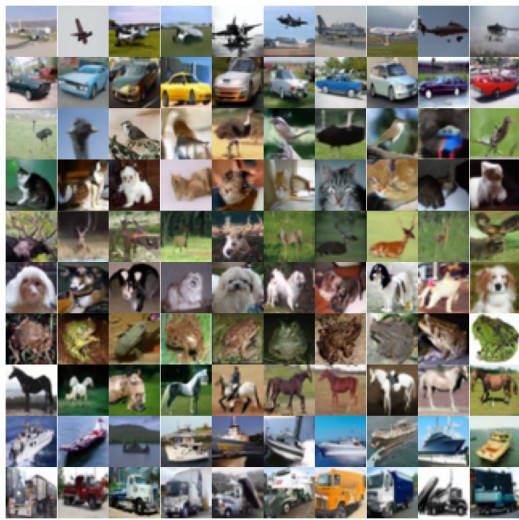

Figure 6: **Samples from the conditional CIFAR-10 model.** The figure depicts samples generated using CFG with a parameter of 0.5. Each row corresponds to a different class.

### C.2.2 Unconditional and Conditional CIFAR-10

We trained the model for 500k iterations with a learning rate of $2 \cdot 10^{-4}$. We started with a linear warmup of 5k iterations and reduced the learning rate by a factor of 10 after every 200k iterations. The typical value of the CE loss after convergence was $\sim 4.4$ while the MSE loss was $\sim 0.03$ so we chose to give the CE loss a weight of $0.001$ to maintain values at the same order of magnitude. In addition, we used EMA with a factor of 0.9999, as done in the baseline model.

For the CDM(unif.) model we used the same hyperparametes except for learning rate, which we set to $1 \cdot 10^{-4}$.

The CDM(OT) model was trained with the same hyperparameters, except for the learning rate, which was set to $1 \cdot 10^{-4}$. Additionally, we trained the model without a learning rate schedule for 1M iterations, as the NLL continued to decrease beyond 500k iterations.

### C.3 Compute Resources

#### C.3.1 CelebA $64 \times 64$

Training the model on CelebA $64 \times 64$ takes 108 hours on a server of 4 NVIDIA RTX A6000 48GB GPUs. Sampling 50k images for FID calculation takes 16 hours on the same hardware.

#### C.3.2 Unconditional and Conditional CIFAR-10

Training the model on CIFAR-10 takes 35 hours on a server of 4 NVIDIA RTX A6000 48GB GPUs. Sampling 50k images for FID calculation takes 9 hours with classifier free guidance and 5 hours without classifier free guidance on the same hardware.

### C.4 Data Augmentation

Following the models to which we compared, for all the datasets we normalized the images to the range $[-1, 1]$ and applied random horizontal flip at training.

### C.5 Classifier Free Guidance

We used CFG to sample conditioned examples in Sec. 4.2 following [18]. We trained our own conditional models on the CIFAR-10 dataset, both for DDM and for CDM, and used label dropout of

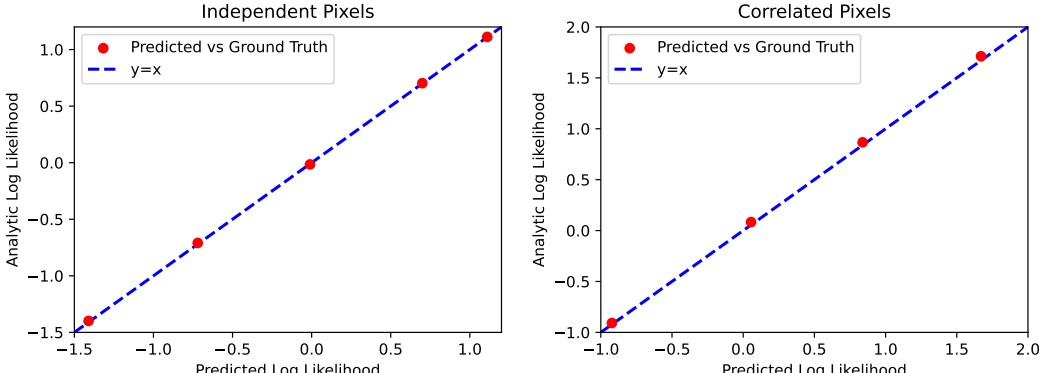

Figure 7: **Log likelihood computation using CDM for toy problems possessing closed form expressions**. The left subplot corresponds to the densities from Sec. D.1, while the right subplot corresponds to the densities from Sec. D.2. The red points on the graphs correspond to differnet $\gamma$ values. The alignment of these points along the diagonal provides evidence supporting our likelihood estimation as stated in Theorem 3.2.

0.1. We selected the best parameter $w$ for each method using a grid search over $w = 0.25, 0.5, 0.75, 1$. In Fig. 6 we show samples from the conditional model. Each row corresponds to a different class.

### C.6  The Addition of The Timesteps $0$ and $T + 1$

As outlined in Sec. 3, we introduce additional timesteps, corresponding to $\boldsymbol{x}_0$, $\boldsymbol{x}_{T+1}$, which are not present in DDMs. This inclusion is fundamental to our method formulation. Importantly, this addition does not alter the number of sampling steps, as we continue initiating the reverse process from t $= T$ and finish it with the denoised result of t $= 1$, following the approach in DDM.

## D  Validating the Log Likelihood Computation on Toy Examples

We validate our efficient likelihood computation by applying it to toy examples with known densities, allowing us to compute the analytical likelihood and verify that our computations align with theoretical expectations.

### D.1  Images With Independent Pixels

We start by experimenting with the uniform distributions $p_\gamma = U\left[-\frac{\gamma}{2}, \frac{\gamma}{2}\right]^{3 \times 32 \times 32}$ with $\gamma \in \{0.25, 0.5, 1, 2, 3\}$. We trained a CDM separately for each of these densities following the procedure outlined in Algorithm 1. These distributions correspond to $32 \times 32$ color images of iid uniform noise. The probability density function (pdf) of the uniform distribution $U\left[-\frac{\gamma}{2}, \frac{\gamma}{2}\right]$ is given by

$$f(x; \gamma) = \begin{cases} \frac{1}{\gamma} & \text{if } -\frac{\gamma}{2} \leq x \leq \frac{\gamma}{2}, \\ 0 & \text{otherwise.} \end{cases} \tag{30}$$

The pdf $p_\gamma(\boldsymbol{x})$ in the $d$-dimensional space is the product of the individual pdfs for each dimension. Since the dimensions are independent, the joint pdf is given by

$$p_\gamma(\boldsymbol{x}) = \prod_{i=1}^{d} f(x_i; \gamma) = \left(\frac{1}{\gamma}\right)^d. \tag{31}$$

The logarithm of the pdf $p_\gamma(\boldsymbol{x})$ is given by $-d \ln(\gamma)$. The log-likelihood is the expectation of $\ln p_\gamma(\boldsymbol{x})$. Since $\ln p_\gamma(\boldsymbol{x})$ is constant, its expectation is that constant. Therefore, the log-likelihood normalized by $d$ is $-\ln(\gamma)$. In the left subplot of Fig. 7 we compare the analytical log likelihood computed using (32) with the estimated log likelihood by our model.

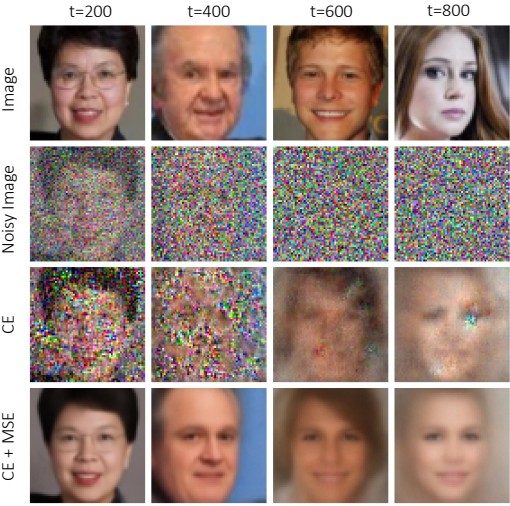

Figure 8: **Comparison of denoising between a model trained using only CE and one trained using both CE and MSE**. We note that the denoising results for the model trained using CE alone are poor, but are better for high noise levels than for lower ones. This resonates with our conclusion that models trained only with CE are only accurate near the real noise level: As denoising with CDM relies on Theorem 3.1, we would expect deteriorating denoising quality the further the real noise level is from $T + 1$, as $f_\theta(\boldsymbol{x})[T + 1]$ is always used for denoising.

## D.2 Images With Correlated Pixels

To further validate our likelihood calculation, we tested it on toy examples of images with correlated pixels. Specifically, we used images of size $3 \times 32 \times 32$ sampled from multivariate Gaussian distributions with $\mu = 0$ and $\Sigma \neq I$. The normalized log likelihood per dimension for a multivariate Gaussian is given by

$$-0.5 \cdot (\ln(2\pi) + 1) - \frac{\ln(|\Sigma|)}{2d}. \tag{32}$$

We defined a sequence of distributions with $\Sigma = \sqrt{1 - \gamma} \, \Sigma_{\text{CIFAR-10}} + \sqrt{\gamma} \, I$, where $\Sigma_{\text{CIFAR-10}}$ is the empirical covariance matrix of the CIFAR-10 dataset, and $\gamma \in \{0, 10^{-5}, 10^{-3}, 10^{-1}\}$. In the right subplot of Fig. 7, we compare the analytical log likelihood computed using (32) with the estimated log likelihood by our model.

## E  More Experiments

### E.1  Analysis of the Effect of Training with Different Losses

Figure 8 provides further qualitative analysis of the effect of training with different losses on the quality of the model's image denoising capabilities. The results illustrate that the model trained using only CE loss achieves poor denoising quality compared to the CDM model trained with both CE and MSE.

### E.2  The Influence of Different Schedulers on the Classifier Performances

First, to assess the classifier's performance, we present the confusion matrices of models trained with and without MSE loss in Fig. 9a. Notably, at lower noise levels, the classifier exhibits high confidence, while at higher noise levels, confidence diminishes. This finding corresponds to the DDPM scheduler [19], which partitions the noise levels to be more concentrated for high timesteps and less concentrated for low timesteps. The similarity between the confusion matrices underlines that CE loss alone is adequate for accurate predictions around the real noise level. However, as demonstrated in the main text, this does not imply that the classifier is optimal in terms of learning the correct logits for any given $t$.

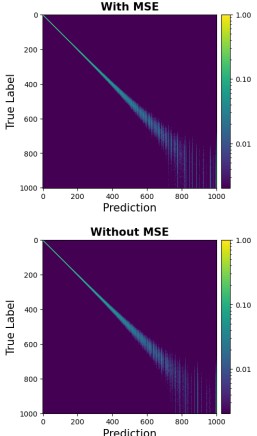

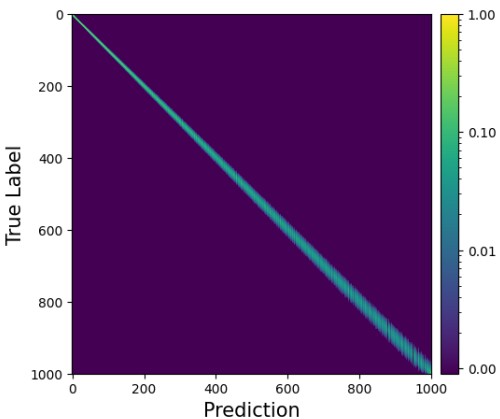

(a) **Confusion matrices evaluated for models trained with both MSE and CE loss (top) and only with CE loss (bottom).** The colors indicate the probabilities. The similarity between the confusion matrices underlines that CE loss alone is adequate for accurate predictions around the real noise level. This is also shown in Fig. (3)

(b) **Confusion matrix CDM(unif.) model evaluated on unconditional CIFAR-10 32 × 32 using the scheduler from TRE**. The colors indicate the probabilities. In contrast to DDPM noise scheduler, with TDR noise scheduler, the classification difficulty is preserved across all timesteps.

In Fig. 9b, we illustrate that the uniform scheduler from [35] induces a uniform difficulty in classification across various noise levels. This is in contrast to the DDM scheduler, depicted in Fig. 9a, in which the classification difficulty increases with the noise level.

