# OpenReview forum: "Classification Diffusion Models: Revitalizing Density Ratio Estimation"
_NeurIPS.cc/2024/Conference — NeurIPS 2024 poster_

### Official Review · Reviewer_fmsZ · 2024-06-26

**Soundness:** 3
**Presentation:** 4
**Contribution:** 4
**Rating:** 8
**Confidence:** 5

**Summary:**

This paper introduces a class of generative models based on neural networks which learn the conditional probability distribution of the noise level given a noisy image. It is trained by combining a maximum-likelihood (cross-entropy) objective with a denoising score matching objective. The resulting generative model can be sampled from by iterative denoising like in diffusion models, and also allows for direct likelihood evaluation (in principle).

**Strengths:**

- The proposed model is original and very interesting. Such an approach has the potential to significantly improve upon diffusion models.

- The experimental results are very convincing (except for NLLs, see below) and show great promise.

- The analysis of the shortcomings of density ratio estimation is also on point and very clear. Although these issues were already evidenced in the previous work of Choi et al., their results were limited to MNIST while CDMs achieve satisfying results on CIFAR10 and CelebA 64x64, and have the added potential benefit of direct likelihood evaluation.

**Weaknesses:**

The biggest weakness of the paper to me is the method used for likelihood evaluation, which has several issues:
- An implicit assumption in the paper is that the marginal distribution of the noise level learned by the model matches the correct one used during training, that is, $p_\theta(t) = p(t)$. Although reasonable, it is very difficult to verify, as it requires marginalizing over the high-dimensional variable $x$.
- There is no guarantee that the resulting density function (eq. (10)) is normalized (i.e., integrates to 1). This can lead to arbitrarily small _reported_ NLLs (potentially even smaller than the entropy of the data). Although the soundness of the approach is verified in a synthetic but high-dimensional example, I think it would be necessary to verify at least this property on a natural image dataset, though this is also very hard to do (but see below).
- There is also no guarantee that the density function in eq. (10) is the density of samples generated by the reverse diffusion (Algorithm 2). CDMs thus contain a priori two models of $p(x)$, one for sampling and one for likelihood evaluation, which could be different. I would also like to see in the final version of the paper (not for the rebuttal) a comparison between the NLLs of these two models on natural images (the latter can be evaluated with an integral along the sampling trajectory using the change of variable formula, as pointed out in the text). If they match, this also verifies that the reported densities are correctly normalized (see previous point).
- Finally, restricting NLL comparisons to "models where likelihood can be evaluated in single shot" seems slightly arbitrary, especially since here sampling is not in a single shot. It would make sense to allow a similar computational complexity for sampling from the model and evaluating the log-density at a data point.

I might be missing subtle details in my assessment though, and I would be happy to update my score if the authors point out misunderstandings.

**Questions:**

- In the proposed approach, $t = T+1$ plays a special role, which seems slightly unsatisfying to me. Having correct log-density estimates using eq. (10) requires an accurate estimation of $\log p(T+1|x)$ for $x$ a clean image, which is very challenging as natural images are very far from typical realizations of Gaussian noise (unless the explicit form of the Gaussian density is enforced here?). However, the MSE loss (which indirectly involves $\nabla_x \log p_\theta(T+1|x)$) seems to fix this problem. Still, is there a way to avoid it altogether with a more symmetric formulation? I liked how the approach of Choi et al. elegantly solved it by considering a path $(x_t)$ of typical realizations of each $p(x|t)$, though I understand that in their setting log-density estimation requires computing a (one-dimensional) integral.

- Out of curiosity, what does Figure 3 look like when the model is trained with MSE loss only? A first observation is that one can add an arbitrary function of $t$ only to the model outputs without changing the MSE loss, so the predicted conditional distributions cannot be right. However, I wonder if this is the only degree of freedom: does the difference between the "true" and predicted $p(t|x)$ depend on $x$? If true, it would show that the CE objective just fixes this constant and is thus not so important.

- I am very puzzled by the DDM denoising result at high noise in Figure 4. Why would they be so bad? Predicting the mean of the dataset is really easy, so this is not a capacity issue but rather a training issue? And why CDM would be specifically better at large noise levels? Is this because $T+1$ plays a special role?

Minor suggestion:
I understand that the "classification" term comes from historical considerations in the density ratio estimation problem. However, it seems ill-adapted here: $t$ is really a continuous variable (which has been discretized, but this does not seem necessary), and therefore $p(t|x)$ is simply a probabilistic model of a conditional distribution that is trained by maximum-likelihood. The title of the paper also sets the incorrect expectation that it is about unifying diffusion models with image classifiers.

**Limitations:**

As mentioned in the discussion, an important aspect that is not studied in this paper is the architecture of the network to approximate $\log p(t|x)$. The community has extensively explored architecture to model scores $\nabla_x \log p(x|t)$, but architectures for energies have been much less studied. As rightfully pointed out, here $f$ is a UNet, so that $\nabla f$ is the gradient of a UNet and thus significantly differs from a UNet (which would be the goal to emulate score-based approaches). Let me suggest a few references that consider this problem:
- Tim Salimans and Jonathan Ho. Should EBMs model the energy or the score? In Energy Based Models Workshop-ICLR, 2021.
- Samuel Hurault, Arthur Leclaire, and Nicolas Papadakis. Gradient Step Denoiser for convergent Plug-and-Play. In International Conference on Learning Representations, 2021.
- Regev Cohen, Yochai Blau, Daniel Freedman, and Ehud Rivlin. It has potential: Gradient-driven denoisers for convergent solutions to inverse problems. In Advances in Neural Information Processing Systems, 2021.

All in all, I very much like this paper, and strongly recommend acceptance. It proposes novel ideas of interest to the generative modeling community, and the strong numerical results demonstrate that this approach has promising potential.

---

> ### Author Rebuttal · Authors · 2024-08-05
>
> We thank the reviewer for the insightful and interesting comments.
>
> **Comparing the one step likelihood computation to an integral along the sampling trajectory**
>
> In theory Eq. (10) should be normalized, but this is a great idea. We’ll definitely add it to the final version. We’d just like to note that while our one-step likelihood computation depends only on the model's output, the integral approach depends also on the sampling method. Specifically, different noise schedulers correspond to different ODE/SDE discretizations, and may affect the quality of the generated samples. This is even if each denoiser by itself is optimal. For a scheduler that leads to low quality samples, the likelihood of the generated images obtained with our one-step approach would be low (high NLL). For example, as we show in Sec. 4.3, we can choose a scheduler that improves the NLL over the test set but leads to a worse FID of the generated samples. We argue that this doesn’t imply the model learned a worse approximation of $p(x)$, but only that the sampling trajectory is worse than that of DDPM.
>
> The DDPM scheduler leads to good generated samples, though. This suggests that the discretization associated with this scheduler leads to an accurate solution of the ODE. And therefore, in this case, we agree that your suggested experiment is interesting and we will include it in our final version. Thanks!
>
> **Likelihood evaluation is performed in a single step, but sampling is not**
>
> We completely agree. However, there may be applications that only require likelihood evaluation for real images, and not sampling of synthetic images. This is the case e.g. in out-of-distribution detection. Therefore, we believe there is merit in separately reporting the NFEs required for likelihood evaluation and for sampling.
>
> The ability of our method to evaluate the likelihood of real images in one step stands in contrast to methods that calculate the likelihood by solving an ODE through a sampling-like process (with repeated evaluations of the network), as in DDMs.
>
> In any case, note from Table 3 that our method is competitive also with methods that evaluate the likelihood using many NFEs.
>
> **The role of t=T+1**
>
> This is a very interesting question. Please note that both in Equation (3) in [34] and in Proposition 1 in Choi’s work [7], the likelihood is extracted from the ratio  $\log\frac{p(x|0)}{p(x|T+1)}$, since $p(x|T+1)$ is known (a Gaussian in this case). In order to achieve a correct likelihood estimation, all the intermediate ratios need to be accurate for the same input $x$, and specifically, the ratios between densities which represent high noise levels need to be accurate when the input is a clean image. For this reason, DRE methods failed to learn distributions of datasets which are more complicated than MNIST. When you incorporate the MSE loss using our theorem, it enforces the log ratio $\log\frac{p(t|x)}{p(T+1|x)}$ to be accurate for any $t$, which solves this problem. However, it would be very interesting in future work to try solving it in different ways.
>
> **Using only the MSE loss**
>
> When training the model using only the MSE loss, the CE is very high (please see Table 1) and Figure 3 is very noisy, which indicates that the model fails to learn the distribution $p(t|x)$ in this case. Your observation is correct and very interesting. The MSE indeed doesn’t change if we add a function of $t$ to the model’s output. This is why the CE loss is important for estimating $p(t|x)$. Without the CE loss, the model can function as a denoiser but is useless for the purpose of outputting the likelihood in a single step. This indeed seems to be the only degree of freedom that the CE loss fixes because the MMSE predictor corresponds to $\nabla_x \log p(t|x)$, which uniquely defines $\log p(t|x)$ up to an additive constant that may depend on $t$.
>
> Note that the difference between the true and predicted $\log p(t|x)$ may depend on $x$. However, in this case it is the same function of $x$ for all $t$. This is true because this function should cancel out after the softmax and also when we take the difference between $\log p(T+1|x)$ and $\log p(t|x)$ in Equation (8).
>
> To conclude, the MSE loss alone is invariant to adding a function of $t$ to the model’s output, and the CE loss removes this degree of freedom. In addition, both losses are invariant to adding a function of $x$ to the model’s output, but the softmax removes this degree of freedom. Importantly, when we calculate the NLL using Equation (10), this function of $x$ is canceled out even before the softmax, since it is added both to $\log p(t|x)$ and $\log p(T+1|x)$.
>
> **Why is the prediction of DDM poor for large $t?$**
>
> We don’t know why diffusion models fail here, and it would be very interesting to explore it in the future. However, we do know why CDM is particularly good at the higher noise levels. Please note that in Theorem 3.1, we show that the optimal denoiser depends on the model's prediction for $p(t|x)$ and $p(T+1|x)$. At the high noise levels, $t$ is close to $T+1$, and as we show in Fig. 3, in this case the model’s prediction for $p(t|x)$ and $p(T+1|x)$ is accurate even without incorporating the MSE loss. This accuracy is further improved when we include the MSE loss.

---

> ### Comment · Reviewer_fmsZ · 2024-08-07
>
> I am happy that you find my suggestions helpful, and that they might improve the paper. I reiterate that this paper should be accepted for the reasons outlined above.
>
> **t = T+1**
>
> In Choi et al., I was referring to what the authors of this paper call the "pathwise method" (section 5.2). The resulting integral has the property that the (infinitesimal) density ratios (as well as the scores) need only be accurate on typical samples from $p(x|t)$ for their respective timestep $t$. Although it seems that they evaluate this more sophisticated method only on Gaussian data.
>
> **Using only the MSE loss**
>
> The reason I asked this question is because this statement is deceptive: "$\nabla \log p$ uniquely defines $\log p$ up to an additive constant". This is true in a strict sense, but not in an approximate sense. In particular, one might have a good _approximation_ of the score _on the support_ of $p$ without having a good approximation of the energy on said support, _when the support is disconnected_. E.g., for a mixture of two well-separated Gaussians, the relative probability of each mode is not captured by the score (it is encoded in the value of the score in-between the two modes, where there is no training data). This is known as the lack of a Poincaré inequality in the maths literature. So this means that in practice, only using the MSE loss might lead to estimated values of $\log p(t|x)$ whose error also depends on $x$.

---

> > ### Author Response · Authors · 2024-08-08
> >
> > **$t=T+1$**
> >
> > Thanks for the clarification. We indeed missed the "pathwise" variant in Choi's paper. This is a clever idea, and we'll definitely mention this in our final version. But, as you note, this method was illustrated only on Gaussian distributions, which suggests that it still fails to solve the problem for complex high-dimensional distributions. This is probably due to the accumulation of many small errors along the path, which our method avoids by the incorporation of the MSE loss.
> >
> > **MSE**
> >
> > Thanks again for the clarification. We absolutely agree. In practice, learning the score by itself can lead to errors in the likelihood, which depend on $x$ (especially for multimodal distributions with well separated modes). In those cases, the CE loss may indeed help fix those errors. We'll clarify this in the final version.

---

### Official Review · Reviewer_93Vh · 2024-07-05

**Soundness:** 4
**Presentation:** 4
**Contribution:** 3
**Rating:** 8
**Confidence:** 4

**Summary:**

A common approach for likelihood estimation is the density ratio estimation (DRE). DRE is a method of modeling the density of a target distribution (or data distribution) in the form of a density ratio with a known reference distribution. The standard normal distribution is often chosen as the reference. To train the density ratio models, a classification-based method called noise contrastive estimation (NCE) is generally used.

However, in high dimensions, the difference between the reference and the data distribution is large, making the classification problem very easy. Consequently, the model does not learn accurate density ratios. To overcome this saturation problem, some have proposed to create intermediate distributions between the target and reference, similar to diffusion models. For example, the transporting ratio estimation (TRE) first creates a bridge between the target and the reference distribution through their linear combinations. The density ratio between two adjacent timesteps—$t$ and $t+1$-th timestep—is trained using the NCE method. However, despite the diffusion-like modifications, TRE has failed to resolve the saturation problem in high-dimensional data.

Therefore, to overcome the limitation of the previous approaches and estimate the log-likelihood efficiently for the high-dimensional datasets, the paper proposes a novel DRE-based diffusion-based generative model. Most importantly, the paper proposes to predict the timestep $t$ for a given observation $x$—$p(t | x)$—when $x$ is sampled uniformly from all intermediate distributions in the bridge. It means that the model predicts how noisy the given input is, unlike the classification problems proposed in TRE. The paper highlights two important benefits of the proposed method.

First, the authors draw the analytical connection between the noise-predicting classifiers and the Bayes-optimal denoisers at each time step. This connection means conventional parameterizations in diffusion models, such as epsilon/score/denoiser, can also be written by the noise-predicting classifiers. Thus, the noise-predicting classifiers can be trained via denoising score matching (DSM) as well as the cross-entropy loss. This training overcomes the saturation problems that originated from classification-based training.

Second, even compared to the TRE, regardless of its scalability issue, the noise-predicting classifiers provide efficiency at the log-likelihood evaluation. The proposed method can estimate the log-likelihood of the (clean) data in one-g, while the TRE requires $T$-number of model evaluations.

In addition, the paper mentions a concurrent work that proposes a similar noise-predicting classifier, but the authors emphasize that the concurrent work didn't incorporate the DSM loss, which is the key ingredient of the proposed approach's scalability.

Finally, the authors demonstrate the proposed method's effectiveness via various experiments, including generative modeling of image generation benchmark datasets.


------------------------
Update the rating from 7 to 8 after the authors' rebuttal

**Strengths:**

One of the paper's key contributions is the introduction of a novel solution for likelihood estimation. The proposed method addresses significant challenges in the density ratio estimation and offers a fresh perspective on the parameterizations in diffusion-based models.

Moreover, the authors effectively motivate the use of DSM by establishing the connection between the time-prediction classifiers and the conventional parameterizations. This training overcomes the limitations of previous DRE-based approaches.

Furthermore, the proposed method exhibits comparable performance and scalability compared to popular diffusion-based generative models, further underscoring the practical effectiveness of the proposed DRE-based method.

**Weaknesses:**

While the proposed method's originality and novelty are clear, its significance is still doubtful. While achieving efficient likelihood estimation is the primary motivation, the paper doesn’t provide sufficient discussion about it—instead, it focuses on demonstrating the improved generation performance of the proposed DRE-based method. Thus, the proposed method would be much more impactful if the authors could demonstrate what becomes available when the likelihood estimation for high-dimensional data works.

Additional discussion about the importance of the classification loss may seem required. It is clear that the time-prediction classifier links to the conventional denoiser parameterization, and thus, this connection equips the use of DSM to train the classifiers, which enables the proposed model's applicability to high-dimensional data. However, this also implies the possibility of training the proposed models without any classification objective.

**Questions:**

N/A

---

> ### Author Rebuttal · Authors · 2024-08-05
>
> **Likelihood estimation is the primary motivation; the paper lacks illustrations of its use**
>
> Our primary motivation was to analyze why existing DRE methods fail to capture distributions of complex high-dimensional data, and to develop theory and a practical DRE method that doesn’t suffer from this limitation. The single-step likelihood evaluation is actually a by-product of our analysis. Nevertheless, we agree with the reviewer that it would be interesting to explore the single step likelihood computation in the context of downstream tasks. We see this as an intriguing direction for future work. For example [R2] showed that a generative model which is able to calculate the likelihood, can be used for zero-shot classification. However, in their case they need to perform many NFEs in order to calculate the likelihood. These applications are out of scope for our paper, but we believe that our method has the potential to be used for solving these tasks in future work.
>
>
>
> **Additional discussion about the importance of the classification loss may seem required**
>
> That's an important point. As shown in Table 1, when training using only MSE loss, the CE is very high. In this case, we can use the model as a denoiser, but it will not function as a noise level classifier and specifically will be useless for calculating the likelihood of a given input using Theorem 3.2. The explanation is that the MSE is invariant to an addition of a function of $t$ to the classifier’s output (since when we take the derivative with respect to $x$ in Eq. 8, it zeros out). Therefore, we must combine the CE loss to remove this degree of freedom.
>
> [R2] Zimmermann, Roland S., et al. Score-based generative classifiers. In: NeurIPS Workshops ‘21.‏

---

### Official Review · Reviewer_ftzp · 2024-07-07

**Soundness:** 3
**Presentation:** 3
**Contribution:** 3
**Rating:** 6
**Confidence:** 5

**Summary:**

This paper introduces a clever connection between Density Ratio Estimation (DRE) and Diffusion Models (DM), showing that optimal denoisers are also optimal noise classifiers. Doing so allows them to construct a new type of loss based on noise classification. This allows DRE methods to inherit the benefits of diffusion models, like strong sampling, while adding new capabilities like single step likelihood estimation.

**Strengths:**

- The mathematical connection is clever and elegant.
- I liked the framing / connections to DRE literature.
- The single step likelihood calculation gives a distinguishing feature that sets the method apart from diffusion models.
- Reasonable ablations were considered, and clear and readable notation was used.
- The method requires an additional backward pass for each denoising/sampling step. In the spirit of the NeurIPS guidelines, I count this clearly labeled limitation as a strength. It's intellectually interesting and also clearly distinguishes that the computation done by their model is different from regular diffusion models in a non-trivial way.

**Weaknesses:**

The proposed approach is functionally equivalent to a diffusion model (using Eq. 8 and with necessary incorporation of MSE loss that matches typical diffusion models), so you might hope to see that you can substantially improve on diffusion models by adding the CE loss as a kind of regularizer. But there are several reasons why it isn't clear that this gives a susbtantial improvement.
- The architecture has to be changed. There's no obvious way to get around that, but it does marginally affect the ease of use for these ideas, and the directness of comparisons (though on this point I agree with the authors' statement that they chose a minimal intervention).
- MSE improvements (Fig. 4) are only marginally different in absolute terms, which is what matters for log likelihood. See some questions about this result.
- The authors only used / compared with relatively old diffusion models, so it's not clear if modest benefits to MSE or FID in some cases would still be seen in more SOTA models. I would expect much more comprehensive experiments to support the idea that using CE loss can improve sampling (Table 2).


The strongest claim of improvement is the single step likelihood calculation. However, there are three weaknesses associated with this result.
Table 3 is a bit of a dubious comparison for two reasons.
- Unlike the ELBO, it's not clear if the proposed estimator gives an upper bound on NLL. I suspect it doesn't. If there are errors in p(t|x), then because it shows up with both signs in Eq. 10, it could contribute to the error in either direction. So we are comparing bounds on NLL to an estimate of NLL which could look lower due only to error.
- The NFE comparisons are also a bit dubious. Function evaluation complexity varies wildly between methods. We should have a fair comparison to the associated diffusion model with the same architecture, but it wasn't clear which one that is in the table. Though it's a bit moot as I wouldn't trust the result anyway, as we don't have a bound here.
- A compelling application of the single step log likelihood evaluation was not discussed. If the log likelihood had been used and evaluated for some downstream application, it might have assuaged worries about the accuracy of the estimate.


Minor comments:
- I didn't like the abstract sentence "directly output likelihood...lacking in most generative techniques", since most common generative techniques today do output likelihood. You clarify later what you mean by "directly" so it is Ok. I just wanted to say that it was off-putting on first read.

**Questions:**

I was really surprised by the fact that your architecture has no timestep conditioning, but matches or outperforms DDMs that do have timestep conditioning in denoising MSE. Have I understood that correctly (i.e. that the DDM in Fig. 4 does have timestep conditioning?) If so, that seems like a really interesting conclusion for diffusion research, as the final appearance of the timestep only shows up in a simple way in Eq. 8, potentially leading to a simplification of other architectures. It would be interesting to see if this effect persists for stronger architectures. On the other hand, maybe the right way to interpret this result is that your architecture essentially outputs the result for all time-steps at once. Eq. 8 selects the correct one gets a score through backprop. Still, that may be an interesting possibility to try to explore for standard diffusion models.

In Fig. 3, I can understand why the classifier is limited, but it wasn't totally clear to me why MSE helps. But there may not be an easy answer to that (besides the superficial one that it is a regularizer of some sort).

**Limitations:**

Limitations were well addressed.

---

> ### Author Rebuttal · Authors · 2024-08-05
>
> **The architecture has to be changed**
>
> We agree with the reviewer that the architecture can't be exactly the same since DDM is a denoiser and CDM is a classifier, which may impact the comparisons a little bit. However, as the reviewer notes, the change in the architecture is minor – we replaced only the last layer of the DDM architecture by a convolution, followed by max-pooling and a linear layer. This has a negligible effect on the number of parameters. It is important to note that this architecture was optimized for DDPM and not for CDM, yet CDM still achieves comparable or better results with it. As we mentioned in the text, we believe that an important and interesting future direction would be to optimize the architecture for CDM.
>
> **Old diffusion models**
>
> We want to emphasize that our main goal is not improving denoising diffusion models, but rather making DRE-based methods work for the first time on datasets more challenging than MNIST. Therefore, we mostly focus on analyzing the reasons that DRE-based methods failed, and show how our theorem leads to a practical algorithm that overcomes these issues. Moreover, we used the same architecture of DDPM only for a fair comparison. In future work, it would be interesting to investigate different architectures which are more suitable for CDM (and in general may be different than the standard denoiser architectures).
>
>
>
> **Upper bound on NLL**
>
> Our method indeed doesn’t upper bound the NLL but rather estimates it. Yet, this is true for all methods in Table 3, which calculate the likelihood and not the ELBO. For example, DDPM-SDE and DDPM++ (which estimate the likelihood and not ELBO) numerically solve an ODE in which each step can accumulate errors.
>
> As for the fact that $p(t|x)$ appears with both signs, this is a key problem with existing DRE based methods, which our method solves. Without incorporating the MSE loss, the model’s prediction is inaccurate for $t$ values that are far from the real $t$ as we show in Fig. 3. Specifically, when calculating the likelihood of a clean image $x$, the model needs to be accurate for both $p(0|x)$ and $p(T+1|x)$. This is the reason that DRE-based methods failed to date. In our case, the addition of the MSE loss solves this problem and makes the NLL calculation more accurate (see also the answer to the last question below). In App. D we show that our NLL calculation is exact on high-dimensional toy examples for which we can calculate the ground-truth NLL analytically.
>
> **Function evaluation complexity varies wildly between methods**
>
> While the evaluation complexity depends on the architecture, in all cases shown in the table, the number of NFEs is at least 100 and in some cases even thousands. In our case, the calculation requires only a single NFE which is much cheaper computationally. In addition, we use the same architecture as DDPM which appears in the table and requires approximately 200 NFEs, and as we show, we achieve a better NLL.
>
> **Validating the log likelihood evaluation on a downstream task**
>
> We agree with the reviewer that it would be interesting to validate the effectiveness of our single step likelihood computation in the context of downstream tasks. We see the use of our method in applications as an intriguing direction for future work. For example [R1] showed that a generative model which is able to calculate the likelihood, can be used for zero-shot classification. However, they need to perform many NFEs in order to calculate the likelihood, whereas our method can do this with a single NFE.
>
> Following the reviewer’s concern, to further validate our NLL evaluation, we will add to the final version more high-dimensional distribution examples (in addition to those already appearing in App. D), for which we can compute the ground-truth NLL analytically.
>
> **The architecture has no timestep conditioning**
>
> Exactly; you understood it correctly. Our theorem suggests that, given the optimal classifier that predicts the timestep (and therefore can't be conditioned on it), we can extract the optimal MMSE denoiser. This effect should persist independently of the architecture and specifically for stronger architectures. It is a unique property of CDMs that doesn't exist in DDMs, since in CDMs the time condition implicitly appears by taking the corresponding classifier's output (i.e., if we want to be conditioned on timestep $t$, we will take the derivative with respect to the $t$-th entry of the classifier outputs, as shown in Theorem 3.1).
>
> Your interpretation that our architecture essentially outputs the result for all time-steps at once is correct, but note that it does so in a very efficient manner. Specifically, if we wanted to do so in a diffusion model, it would have to output $T$ predicted noise maps at once. Our model, on the other hand, only outputs $T$ scalars. It is the gradient of each output that gives the predicted noise map.
>
>
>
>  **Why does MSE help in Fig. 3?**
>
> That is a very important point. When training the model using only the CE loss, the model’s prediction $p(t|x)$ may be accurate for the correct noise level but not for more distant noise levels (as shown in Fig. 3). This is why DRE-based methods have failed to capture the distribution of images to date. As Theorem 3.1 shows, the optimal denoiser for timestep $t$ depends on the model prediction for both $\log p(t|x)$ and $\log p(T+1|x)$. Therefore, when we add the MSE loss on the derivative of the difference between them (using the formula from Theorem 3.1), we enforce the classifier’s output to be accurate not only in it’s $t$-th entry (as can be achieved using only CE) but also in its $(T+1)$-th entry. By enforcing this, we obtain an optimal classifier that predicts the correct probability $p(t|x)$ both locally around $t$ (thanks to the CE) and globally for distant $t$ values (thanks to the MSE).
>
>
>
> [R1] Zimmermann, Roland S., et al. Score-based generative classifiers, NeurIPS Workshops ‘21.‏

---

> > ### Comment · Reviewer_ftzp · 2024-08-09
> > **Response to rebuttal**
> >
> > Thank you for the responses.
> >
> > I will maintain my score - I think this is nice theoretical foundation work but as "the main goal is not improving [the state of the art models], but rather making DRE-based methods work for the first time on datasets more challenging than MNIST", it somewhat limits the impact.
> >
> > FYI, while bidding for papers I noticed a concurrent similar work that you may want to check out/cite as concurrent: "diffusion models as noise classifiers" or something similar.

---

> > > ### Author Response · Authors · 2024-08-09
> > >
> > > **Main goal**
> > >
> > > Thanks. We want to emphasize that although improving diffusion models wasn't our main motivation, we succeeded in achieving comparable or better results than the base models using the same architectures (which are optimized for denoising and not for our task). As we mentioned, an important direction for future work would be to design architectures that are optimized for CDM and to scale our work to larger datasets. Given our preliminary results, we believe that CDM has the potential to improve upon diffusion models in large scale challenging settings.
> > >
> > > **Concurrent work**
> > >
> > > Thanks for letting us know. We'll look it up and cite accordingly.

---

### Official Review · Reviewer_15GS · 2024-07-13

**Soundness:** 3
**Presentation:** 3
**Contribution:** 3
**Rating:** 5
**Confidence:** 2

**Summary:**

This work develops a new generative framework called the classification diffusion models (CDMs) based on the density ratio estimation (DRE), by establishing an interesting connection between the DDPM's denoiser and noise-predictive classifier, which also helps the exact likelihood computation in a single pass. As is claimed by authors, the proposed method is the first DRE-based technique to successfully generate images of the CIFAR-10 dataset.

**Strengths:**

1. This work is well-written and clearly formulated.
2. The proposed algorithm is theoretically grounded.
3. The experiments are convincing, which verify the effectiveness of proposed methods.

**Weaknesses:**

1. The main concern is the capability or potential of DRE-based methods to successfully learn data distributions, particularly when applied to datasets with large scales in practice. As is claimed by authors, the current development of DRE-based modeling is up to the CIFAR-10 dataset (MNIST before), which is too small to indicate the validity compared to more standard methods (e.g. DDPM).
2. Another major concern is that CDMs can be more computationally challenging than DDPMs in sampling. According to Algorithm 2, CDMs require an extra backward propagation (BP) than DDPMs for each time step when denoising, which is quite expensive.

**Questions:**

1. Please provide more details of questions raised in the weaknesses section above.
2. Check minor grammar typos, e.g. in Line 258, " trains a classifiers...".

**Limitations:**

As is stated by authors, it is worthy to explore BP-friendly architectures to alleviate the computation inefficiency of CDMs.

---

> ### Author Rebuttal · Authors · 2024-08-05
>
> **The potential of DRE-based methods to learn data distributions of larger scale datasets**
>
> We acknowledge that the Celeb-A 64x64 and CIFAR-10 32x32 datasets we experimented with are not very large and high-dimensional by today’s standards, and we certainly agree that extending the evaluation to datasets like ImageNet would be valuable. Unfortunately, however, we lack the computational resources to do so. For instance, as reported in [9], Table 10, achieving satisfactory results (in terms of FID) on ImageNet 128x128 requires a large amount of computation, reaching roughly 521 days on a single V100 GPU. We can run on eight A6000 GPUs, with which we approximate it will take us 56 days to train a standard DDM with the same architecture and number of iterations, using a batch size which is smaller by a factor of 2.
>
> Nevertheless, we believe that the preliminary exploration we present on CIFAR-10 and Celeb-A is still valuable for the research community, as it analyzes why DRE-based methods have failed to capture distributions of high-dimensional data to date, and suggests a practical algorithm based on our theoretical result for overcoming this problem. Furthermore, we truly believe that with sufficient computational resources, future works will be able to train our method on larger-scale datasets.
>
> **CDMs can be more computationally challenging than DDPMs in sampling**
>
> We agree with the reviewer and have mentioned it as a limitation in the main text. Indeed, while a DDM requires a single forward pass for each denoising step, a CDM requires both a forward pass and a backward pass. Nevertheless, the computational cost of performing a forward and a backward pass through a network depends on its architecture. In this work, we chose to use the same architecture as that used by DDPM [19] to isolate the impact of our algorithmic approach from the choice of the model architecture when comparing it to DDMs. However, it is an important future direction to explore architectures that are particularly optimized for CDMs and that alleviate the gap in computational complexity. Such architectures should have the property that performing a forward pass and a backward pass through them is computationally similar to performing only a forward pass in a regular DDM. This could potentially be achieved e.g., by relying only on the encoder part of the UNet. However, we leave this exploration for future work.
>
>
> **Minor grammar typos**
>
> Thanks. We will fix them in the final version.

---

> > ### Comment · Reviewer_15GS · 2024-08-12
> >
> > Thanks for the reply! I thinks the connection between the DDPM's denoiser and noise-predictive classifier is interesting, but the large-scale effect and inefficiency concern still holds. I will keep the score.

---

### Decision · Program_Chairs · 2024-09-25

**Decision:**

Accept (poster)

**Comment:**

This paper focuses on diffusion-based generative models using density ratio estimation along with a classifier that predicts the level of noise that has been added to a data point in a TRE-like manner (transporting ratio estimation). By deriving analytical connections, the paper shows how to train noise-predicting classifiers using denoising score matching with the cross-entropy loss. The benefit is that log-likelihood of data can be estimated in one function evaluation.

The reviewers found the approach original and interesting enough to warrant publication at NeurIPS. The major contribution is a novel way to think about both density ratio estimation and diffusion models which the reviewers found to have a lot of potential.

While these analytical connections were satisfying to reviewers, the generative performance of the method lags behind regular diffusion model approaches. Hence, the main benefit is to advance density-ratio methods, but this is not the most performant or promising class of methods, and is not the highest bar to compare against. Hence I find acceptance as a poster to be most appropriate.